# A paralog of Pcc1 is the fifth core subunit of the KEOPS tRNA-modifying complex in Archaea

Marie-Claire Daugeron[1,5], Sophia Missoury[1,3,5], Violette Da Cunha [1,4], Noureddine Lazar[1], Bruno Collinet [1,2], Herman van Tilbeurgh [1] ✉ & Tamara Basta [1] ✉

In Archaea and Eukaryotes, the synthesis of a universal tRNA modification, $N^6$-threonyl-carbamoyl adenosine ($t^6A$), is catalyzed by the KEOPS complex composed of Kae1, Bud32, Cgi121, and Pcc1. A fifth subunit, Gon7, is found only in Fungi and Metazoa. Here, we identify and characterize a fifth KEOPS subunit in Archaea. This protein, dubbed Pcc2, is a paralog of Pcc1 and is widely conserved in Archaea. Pcc1 and Pcc2 form a heterodimer in solution, and show modest sequence conservation but very high structural similarity. The five-subunit archaeal KEOPS does not form dimers but retains robust tRNA binding and $t^6A$ synthetic activity. Pcc2 can substitute for Pcc1 but the resulting KEOPS complex is inactive, suggesting a distinct function for the two paralogs. Comparative sequence and structure analyses point to a possible evolutionary link between archaeal Pcc2 and eukaryotic Gon7. Our work indicates that Pcc2 regulates the oligomeric state of the KEOPS complex, a feature that seems to be conserved from Archaea to Eukaryotes.

The genes that are shared by all cellular life forms are coding for a very restricted set of about 60 proteins, primarily involved in translation[1,2]. Two protein families in this set, Sua5/TsaC (COG0009) and Kae1/TsaD/Qri7 (COG0533), are tRNA-modifying enzymes involved in the synthesis of a noncanonical nucleotide $N^6$-threonyl-carbamoyl adenosine ($t^6A$)[3–5]. This modified adenosine derivative is always located at position 37, adjacent to the anticodon, in almost every tRNA reading A-starting codons (ANN, N being one of the four canonical nucleotides A, C, G, or U)[6–8]. $t^6A$ is critical for accurate decoding of mRNA[5,9–13] and for binding of tRNA to the ribosome and translocation to the P site during protein synthesis[14–16]. Mutations in the $t^6A$ biosynthetic genes can be lethal in bacteria and archaea[5,17] or result in a wide range of severe phenotypes in eukaryotes such as translation defects[18] mitochondrial dysfunction[19] genomic instability[20,21], transcription defects[22,23], telomere shortening[24,25] and body development defects[26].

In humans, such mutations were linked to a rare genetic disease, Galloway-Mowat syndrome (GAMOS), whereby neurological and renal functions are severely affected and result most often with childhood mortality[27,28].

The biosynthesis of $t^6A$ is a two-step process in which Sua5/TsaC proteins first catalyze the formation of threonyl-carbamoyl adenylate (TC-AMP)[29]. This unstable intermediate is used as substrate by the Kae1/TsaD/Qri7 enzyme family which catalyzes the transfer of a threonyl-carbamoyl group to the $A_{37}$ of the tRNA substrate[30,31] (Fig. 1A). While the mitochondrial ortholog Qri7 is a standalone enzyme acting as a homodimer, in Bacteria the TsaD protein associates with two bacteria-specific proteins, TsaB (an inactive paralog of TsaD) and TsaE (a P-loop ATPase) into DEZ complex[32]. In archaea and eukaryotes, the Kae1 enzymes are part of a multiprotein complex named KEOPS (Kinase Endopeptidase and Other Proteins of Small size)[24] or EKC

[1]Université Paris-Saclay, CEA, CNRS, Institute for Integrative Biology of the Cell (I2BC), 91198 Gif-sur-Yvette, France. [2]Institut de Minéralogie de Physique des Matériaux et de Cosmochimie (IMPMC), Sorbonne-Université, UMR7590 CNRS, MNHN, Paris, France. [3]Present address: Department of structural biology and chemistry, Institut Pasteur, Paris, France. [4]Present address: Génomique Métabolique, Genoscope, Institut François Jacob, CEA, CNRS, Univ Evry, Université Paris-Saclay, 91057 Evry, France. [5]These authors contributed equally: Marie-Claire Daugeron, Sophia Missoury. ✉e-mail: herman.van-tilbeurgh@i2bc.paris-saclay.fr; tamara.basta@i2bc.paris-saclay.fr

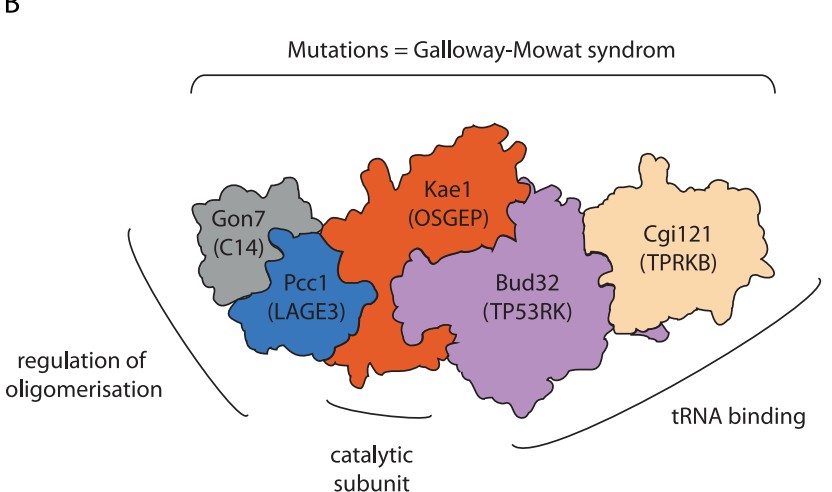

**Fig. 1 | KEOPS complex structure and function in t⁶A biosynthesis. A** Enzymatic synthesis of t⁶A. Sua5/TsaC protein family catalyzes the condensation of L-threonine, bicarbonate or CO₂ and one ATP molecule to form an unstable intermediate threonylcarbamoyl-adenylate (TC-AMP). Kae1/TsaD/Qri7 family catalyzes the transfer of the threonylcarbamoyl moiety of TC-AMP to adenosine 37 within the anticodon loop to form t⁶A modified tRNA. **B** Schematic representation of the KEOPS structure and functional assignment of subunits. The complex is a linear assembly of five subunits Cgi121/Bud32/Kae1/Pcc1/Gon7. The names of the human orthologs are given in the brackets. Gon7/C14 subunit has been found only in fungi and animals. The shape of each subunit roughly recapitulates the corresponding crystal structures in Gon7-Pcc1 (PDB 4WXA), Pcc1-Kae1 (PDB 3ENO), Kae1-Bud32 (PDB 3EN9), and Bud32-Cgi121 (PDB 4WWA) binary complexes. Genetic and in vitro functional assays established that Bud32 and Cgi121 function in tRNA recognition and binding. Pcc1 forms homodimers and provides the interface for oligomerization of the four subunit (4SU) KEOPS complex into a dimer of two heterotetramers. In fungi and animals, the Gon7/C14 subunit binds to Pcc1 through its dimerization interface and blocks the assembly of the KEOPS dimer. Mutations in all five genes encoding human KEOPS were linked with a severe genetic disease called Galloway-Mowat syndrome.

(Endopeptidase-like Kinase Chromatin-associated)[22]. For simplicity, we will use the name KEOPS thereafter. For a detailed account on KEOPS structure and function the reader is referred to a recent review[33]. KEOPS was initially isolated from yeast by tandem affinity purification, and it contained Kae1, a metalloprotein from the ASHKA superfamily, Bud32 a Rio-like protein kinase and three small proteins Pcc1, Cgi121 and Gon7 of unknown function. The latter protein is specific for fungi, while Bud32, Pcc1 and Cgi121 are conserved both in archaea and eukaryotes[22,24]. The complete structure of the KEOPS complex could be modelled based on crystal structures of subcomplexes[34–36]. The complex exhibits a linear arrangement of subunits in the order Pcc1-Kae1-Bud32-Cgi121. The fifth subunit, Gon7, binds to Pcc1 on the opposite side from Kae1 in human and yeast KEOPS[35,37] (Fig. 1B).

Despite considerable progress made in the last decade, how the auxiliary proteins Pcc1, Bud32, Cgi121, and Gon7 emerged and why they were recruited within the KEOPS complex in Archaea and Eukaryotes remains poorly understood. Using in vitro reconstituted KEOPS complex from the archaeon *Pyrococcus abyssi*, we initially showed that PaKae1, PaPcc1, and PaBud32 are necessary and sufficient for the synthesis of t⁶A while the addition of PaCgi121 stimulated the activity[38]. Using EMSA experiments, we showed that the binary Pcc1–Kae1 complex did bind to tRNA while Cgi121 alone did not. Beenstock and colleagues recently showed that Cgi121 recruits tRNA to KEOPS by binding specifically to its 3′ CCA tail. Modelling of the tRNA/KEOPS complex based on low-resolution electron microscopy and biochemical data,

further indicated that the four KEOPS subunits form an extended tRNA-binding surface and are required for correct positioning of the A₃₇ nucleoside in the catalytic site of Kae1[39].

Pcc1 is a small, ~10 kDa globular protein composed of two antiparallel alpha helices and three-stranded beta-sheet. It has some structural resemblance with the K homology (KH) domain, which functions as a single-stranded DNA and RNA binder[34]. The X-ray crystal structure of Pcc1 from *Pyrococcus furiosus* showed that this protein forms homodimers whereby the two protomers are arranged in antiparallel fashion and form a continuous 6-stranded beta-sheet with two alpha helices packed against one side of the sheet[34]. The Pcc1 homodimer provides two equivalent and non-overlapping surfaces for binding Kae1 and is responsible for the dimerization of the archaeal and human KEOPS complexes in vitro with 2:2:2:2 binding stoichiometry[37,38]. Perturbation of Pcc1-mediated dimerization of archaeal KEOPS by mutation of one Kae1-Pcc1 interface had no effect on t⁶A biosynthesis activity in vitro[40]. However, genetic experiments established that the synthetic Pcc1-Pcc1 fusion protein engineered to form homodimers in cis requires two intact Kae1 binding surfaces to support the growth of yeast thus raising the possibility that KEOPS could act as a dimer in vivo[34]. In fungi, Pcc1 interacts with Gon7 which is a small ~14 kDa intrinsically disordered protein that becomes partially structured upon interaction[35]. The specific function of Gon7 is unknown but this protein was shown to be essential for the t⁶A synthesis and telomere maintenance in yeast[34,41]. Gon7 binds to Pcc1 at

the opposite side of Kae1. It has a different fold from Pcc1 and its binding to Pcc1 precludes the dimerization of the yeast KEOPS[35].

Sequence similarity searches failed to detect Gon7 homologs outside fungi but using proteomics approaches the C14ORF142 protein (hereafter C14) was identified as the fifth core subunit of human KEOPS[37,42]. It was further shown that C14 and Gon7 have the same structure and they interact identically with the LAGE3 (the human ortholog of Pcc1) and Pcc1 subunits, respectively. Therefore, C14 and Gon7, despite their weak sequence similarity, are homologs. Like Gon7, C14 binds to the homodimerization interface of LAGE3 and thereby prevents the dimerization of the human KEOPS complex[28]. Mutations in human GON7 lead to a milder form of Galloway-Mowat syndrome. Enzymatic assays using in vitro reconstituted system further established that C14 stimulates the t⁶A synthetic activity up to four-fold[37].

The presence of a homologous fifth core subunit in the KEOPS complex from two distinct eukaryotic lineages raised the possibility that Gon7/C14 orthologs could also be present in Archaea. In search for the putative fifth KEOPS subunit in archaea, we identified in silico a small uncharacterized protein that showed significant sequence similarity to the Pcc1 KEOPS subunit. We present here the biochemical and structural characterization of this protein, named Pcc2, and establish it as a fifth core subunit of the archaeal KEOPS complex. Pcc2 shares several sequence and structural similarities with the eukaryotic Gon7/C14 proteins suggesting that the 5-subunit complex is the ancestral form of the KEOPS complex. Our work lays foundation for the future comprehensive studies of the biological and biochemical function of Pcc2 and warrants searches for the fifth core subunits in eukaryotic lineages outside animals and fungi.

## Results

### A paralog of Pcc1 is widely distributed among archaea

The recent discovery of a fifth subunit of KEOPS in metazoa raised the possibility that a corresponding subunit could exist in archaea. In search for such a subunit, we noticed in the model archaeon *Thermococcus kodakarensis* KOD1 the existence of two genes (TK1253 and TK0642) both annotated as KEOPS subunit Pcc1 in the NCBI database. TK0642 encoded an 82 aa protein (UniProt Q5JF86) while TK1253 encoded an 85 aa protein (UniProt Q5JGMS) (Supplementary Fig. 1A). The two proteins displayed rather low pairwise sequence identity (22%) but high sequence similarity (55%). The alignment of their sequences with the previously described Pcc1 from *Pyrococcus furiosus* (PfPcc1)[34] showed that TK0642 is more similar to PfPcc1 (55% sequence identity versus 29% sequence identity for TK1253) (Supplementary Fig. 1B). This suggested that TK0642 and TK1253 are paralogs derived from a gene duplication event whereby TK0642 encodes the canonical Pcc1 protein. We named the product of the gene TK1253 Pcc2 to underline the evolutionary link with Pcc1 protein family.

Next, we set out to investigate the distribution of Pcc2 among archaea using HMM-based sequence similarity searches. We detected Pcc2 orthologs in 11% of DPANN, 38% of Asgard, 83% of TACK, and 88% of Euryarchaeota genomes. Except for the DPANN superphylum where Pcc2 was found in a single lineage (Altiarchaeota), the vast majority of archaeal genomes contained both Pcc1 and Pcc2 encoding genes (Fig. 2A and Supplementary Fig. 2). These numbers should probably be regarded as conservative since we may have missed highly divergent orthologs. Moreover, when searching for Pcc2 orthologs in specific lineages we found that they were missing in protein sequence databases but we could detect them by performing manual tblastn searches, thus indicating that some *pcc2* genes (and likely also *pcc1* genes) were left undetected by automated gene annotation tools.

The alignment of representative Pcc1 or Pcc2 sequences from 27 archaeal phyla revealed low amount of global sequence conservation within each gene family, except for a few highly conserved residues grouped in two separate regions (box 1 and box 2) (Supplementary Fig. 3). While the box 1 sequence contains conserved residues in both

protein families, the box 2 is specific to each family. Of note, the conserved residues $N^{72}$, $S/T^{73}$ in Pcc1 sequences are replaced by negatively charged residues $D/E^{72}$, $D/E^{73}$ in Pcc2 sequences (*Pyrococcus furiosus* numbering, UniProt Q8TZI1) (Fig. 2B). Consistent with them being paralogs, the phylogenetic tree of representative Pcc1 and Pcc2 sequences showed bipartite topology with one monophyletic clade corresponding to Pcc1 orthologs and the other to Pcc2 orthologs (Fig. 2C).

Collectively, the data show that Pcc2 and Pcc1 are paralogs emerging after an ancient gene duplication event that occurred before the diversification of Archaea. Following the duplication, the proteins were conserved in the majority of archaeal lineages and they diverged in their C-terminal part which harbors paralog-specific conserved residues. The discovery of Pcc2 in Archaea offered the exciting possibility that this paralog evolved to provide a specific function within the KEOPS complex.

### Pcc2 protein co-purifies with the archaeal KEOPS

To investigate if Pcc2 could bind to the KEOPS complex we produced in *E. coli* subcomplexes containing two or three of the known four KEOPS subunits from *Pyrococcus abyssi* (PaPcc1, PaKae1, PaBud32, PaCgi121) plus PaPcc2 whereby only PaPcc1 or PaPcc2 carried a hexahistidine tag (see material and methods for details). The corresponding cell lysates were mixed, and recombinant complexes were purified using classical two-step procedure consisting of Ni-NTA affinity chromatography followed by size exclusion chromatography.

This protocol yielded a complex containing PaCgi121 (C), PaBud32 (B), PaKae1 (K), PaPcc1 (P1), and PaPcc2 (P2) that migrated slightly below PaPcc1 (Fig. 3). We named this 5-subunit (5SU) complex CBKP1P2. After the first step of purification, the CBKP1P2 complex co-eluted with an excess of PaPcc1 and PaPcc2 subunits (Supplementary Fig. 4) which could be separated by size exclusion chromatography. PaPcc1 and PaPcc2 eluted as a single ~29 kDa peak suggesting they could form heterodimers (Supplementary Fig. 4) and that PaPcc1 could be the direct binding partner of PaPcc2 within the CBKP1P2 complex.

Because of their similarity in sequence and size, we hypothesized that PaPcc2 might mimic PaPcc1. To test this, we co-expressed PaCgi121, PaBud32, PaKae1, and PaPcc2His6 and applied the same two-step purification protocol. The three canonical KEOPS subunits co-purified with PaPcc2 thus showing that PaPcc2 can indeed replace PaPcc1 probably by binding directly to Kae1. We named this complex CBKP2 (Fig. 3, Supplementary Fig. 4). Alongside CBKP2, an excess of PaPcc2 could also be purified as a single peak with an apparent molecular mass of ~26 kDa suggesting that PaPcc2 is a homodimer in solution and could, in analogy to Pcc1 homodimer, potentially provide the interface for the dimerization of the CBKP2 complex.

To investigate the oligomeric state of CBKP1P2 and CBKP2 we started by comparing their size exclusion profile with that of CBKP1, known to build dimers in solution. CBKP1 eluted as two separate peaks (66.3 mL and 73.6 mL) both containing the four subunits suggesting that the complex exists as a mixture of dimers and monomers in our experimental conditions (Fig. 3, Supplementary Fig. 4). The CBKP1P2 and CBKP2 each eluted as a single peak but their elution volumes (77.3 mL and 68.9 mL, respectively) were substantially different from each other and from that of the CBKP1 complex suggesting that the three complexes adopt different quaternary structures.

Collectively, these experiments demonstrated that PaPcc2 binds both to CBKP1 and to CBK complexes. The capacity of PaPcc2 to form heterodimers with PaPcc1, suggests direct interaction with PaPcc1 or, in absence of Pcc1, with the catalytic subunit Kae1. Pcc1 and Pcc2, alone or in combination, seem to dictate the oligomeric state of PaKEOPS.

### The Pcc2 crystal structure closely resembles that of Pcc1

Using the pure PaPcc2 fractions we collected during purification of CBKP2 complex we solved the crystal structure of PaPcc2 by

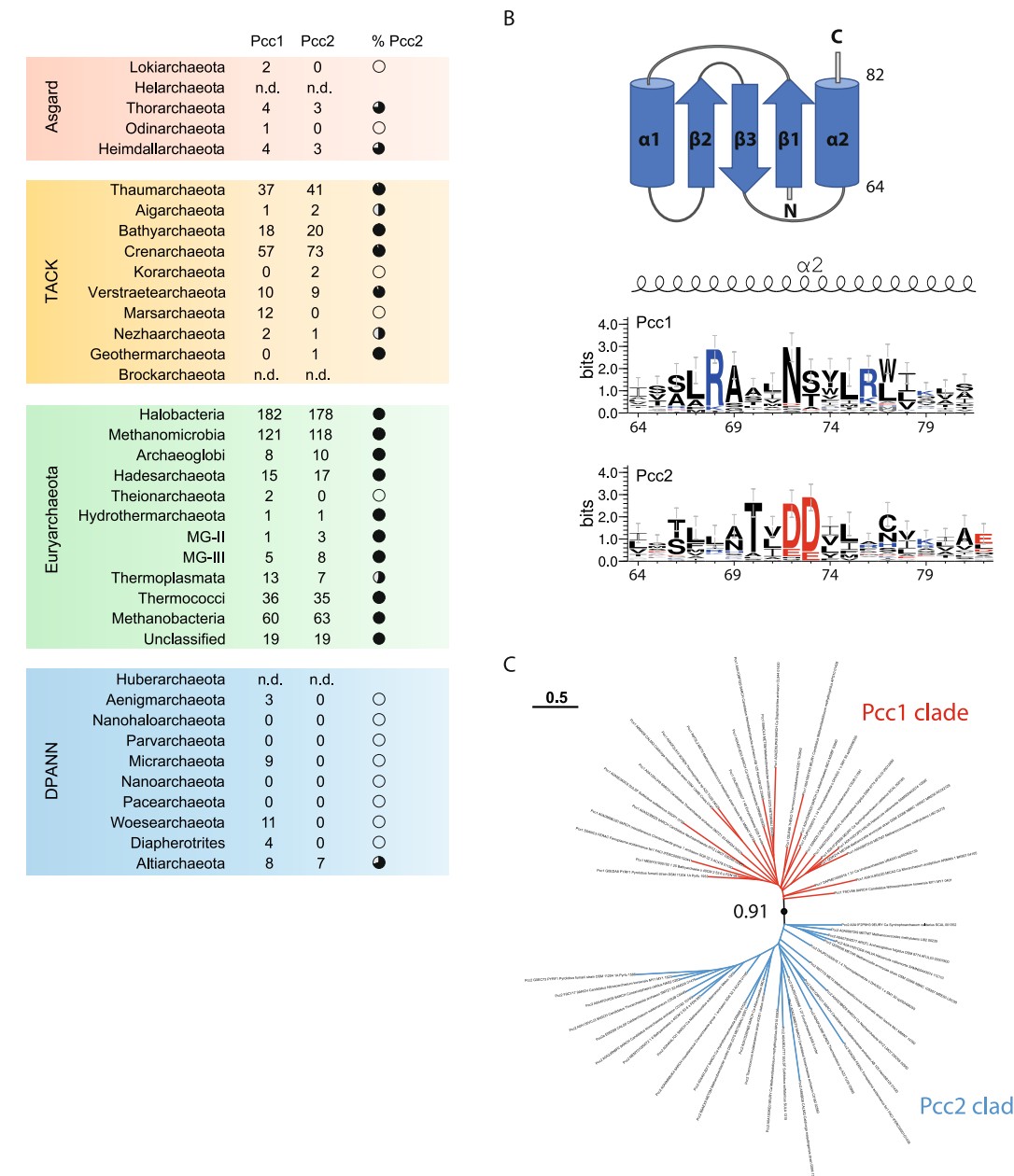

**Fig. 2 | Identification of Pcc2 proteins in archaea. A** Prevalence of Pcc1 and Pcc2 encoding genes among archaeal phyla. Number of identified Pcc1 or Pcc2 proteins is indicated in the corresponding columns. Full black circle corresponds to 100% and white circle to 0% of genomes containing both Pcc1 and Pcc2 encoding genes. n.d. not determined. Source data are provided in Supplementary Data 1. **B** Pcc1 and Pcc2 proteins exhibit divergent C-terminal sequences. The alignment of 27 representative Pcc1 and Pcc2 proteins was used to generate the sequence logo (see Supplementary Fig. 3 for complete logo). Only the part of the sequence corresponding to the α2 helix of Pcc1 proteins is shown. The complete topology of Pcc1 proteins is shown on the top. The size of the letters is proportional to the conservation score. The amino acid numbering is from Pcc1 ortholog of *Pyrococcus furiosus* (UniProt Q8TZI1). **C** Pcc1 or Pcc2 sequences segregate into two distinct clades. Maximum likelihood phylogenetic tree was inferred from an alignment of 27 Pcc1 or Pcc2 sequences representative of the archaeal diversity. The bootstrap value for the branch separating Pcc1 and Pcc2 clades was evaluated from 500 boot trees using transfer bootstrap expectation. Tree scale gives the correspondence between the branch length and sequence evolution in number of substitutions per site. The phylogenetic signal contained in the sequence dataset was insufficient to resolve robustly the topology of the tree within each clade.

molecular replacement at a resolution of 1.84 Å (Supplementary Table 1). The asymmetric unit contained three nearly identical (RMSD around 1.3 Å) PaPcc2 copies. Two copies form a tight dimer whereas the third copy is more loosely associated with the two others. The latter copy forms a tight dimer with a symmetry related partner. PaPcc2 forms a three stranded anti-parallel β-sheet with three α helices packed against one face of the sheet (βαα'ββα topology) (Fig. 4A). Interestingly, despite moderate pairwise sequence identity (29%), PaPcc2 and Pcc1 from *Pyrococcus furiosus*

(PfPcc1) superpose very well (Z score of 7.2 and RMSD of 1.48 Å for 76 equivalent Cα positions) (Fig. 4B). The main structural difference between PfPcc1 and PaPcc2 resides in the connection between strands β1 and β2 that consists of a single α helix (α1) for PfPcc1 and two perpendicularly oriented α helices (α1 and α1') for PaPcc2. Also the PaPcc2 homodimer structure is very similar to that of PfPcc1. In both dimers, the two monomers associate via their β1 strands to form a continuous six-stranded anti-parallel β-sheet and the two main helices are involved in hydrophobic packing. The

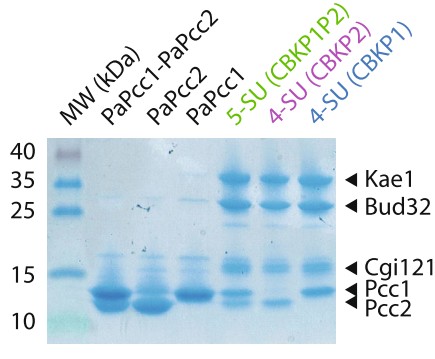

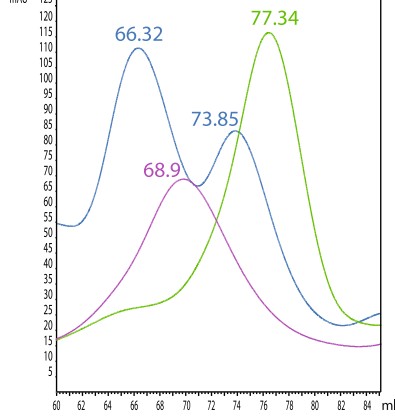

**Fig. 3 | Purification of recombinant PaKEOPS complexes.** SDS-PAGE analysis (left panel) and size-exclusion chromatography profiles (right panel) of purified recombinant KEOPS complexes produced in *E. coli* cells. See Supplementary Fig. 4 for complete chromatograms. The purification experiments were repeated at least four times with reproducible results.

superposition of the two homodimers reveals about 10 Å longitudinal sliding of PaPcc2 protomers alongside β1 strand (Fig. 4B).

In the X-ray crystal structure of archaeal Kae1-Pcc1 complex, the α1 helix of Pcc1 is part of a four-helix bundle building the Kae1-Pcc1 interaction interface[34]. To explore how the observed differences in α1 helix conformation could impact the interaction of PaPcc2 with Kae1 protein we superposed the structures of the PaPcc2 homodimer with the PfPcc1 homodimer in complex with one Kae1 protein from the archaeon *Thermoplasma acidophilum* (Fig. 4C). This revealed a significant displacement of both α1α1′ and α2 helices with respect to their PfPcc1 counterparts suggesting that PaPcc2 likely binds differently to Kae1. This may be one of the reasons why there are considerable differences in the quaternary structures of the CBKP1, CBKP1P2, and CBKP2 complexes (see further below).

## Structure of the PaPcc1-PaPcc2 heterodimer

During the gel-filtration purification step of PaCBKP1P2, we collected an excess of PaPcc1 and PaPcc2 as a single peak suggesting that these two proteins form a heterodimer. However, we could not exclude this peak contained a mixture of homodimers and/or heterodimers and we, therefore, decided to crystallize the protein compound in these fractions. We solved its structure at 3.18 Å resolution, establishing that PaPcc1-PaPcc2 heterodimer was present in the crystal (Supplementary Table 1, Fig. 5A). The two proteins align their respective β1-strands and create a six stranded anti-parallel β-sheet similar to one observed in the PaPcc2 and PfPcc1 homodimers. The heterodimer is further stabilized by the hydrophobic packing of the two C-terminal helices. Overall, the Pcc1 and Pcc2 homodimers and the Pcc1-Pcc2 heterodimer have all very similar structures (Fig. 5B). The structures of PaPcc2 as present in the homodimer and in the PaPcc2-PaPcc1 heterodimer are almost identical (RMSD 0.98 Å) suggesting that binding of PaPcc2 to PaPcc1 does not induce a significant conformational change in the PaPcc2 structure.

PfPcc1 binds to Kae1 by engaging its two helices in a four helical bundle. The opposite surface in principle should remain available to form dimers and hence induce the formation of a (super)dimer of two heterotetramers. Our data show that Pcc2 mimics a Pcc1 subunit upon formation of the heterodimer. Although we do not dispose of the CBKP1P2 structure, in this complex Pcc2 binds very likely to Pcc1 as it does in the heterodimer. The subunit interface (1125.1 Å² vs. 771.9 Å²) and the number of potential hydrogen bonds across interface (18 vs. 8) is much higher in the PaPcc1-PaPcc2 heterodimer than for the PaPcc2 homodimer, suggesting the heterodimer might be the preferred arrangement.

## SAXS analysis of PaKEOPS complexes in solution

To learn more about the oligomeric status of the CBKP1, CBKP1P2, and CBKP2 complexes in solution we performed SAXS experiments. The Guinier plots for the three complexes were linear, testifying that the samples were monodisperse (Supplementary Fig. 5). From these plots, we calculated the radius of gyration, a measure of the spread of molecular mass, which was almost identical for CBKP1 and CBKP2, but was smaller for CBKP1P2 suggesting that the latter had a lower mass. (Supplementary Fig. 5).

We previously established using gel filtration that PaCBKP1 forms superdimers[38]. Our present SAXS data for this complex yield a Mw of approximately 186 kDa, consistent with a 2:2:2:2 superdimer stoichiometry (Supplementary Table 2). Similarly, for CBKP2 we obtained a Mw of approximatively 180 kDa indicating that CBKP2 also forms a superdimer with 2:2:2:2 stoichiometry (Supplementary Table 2). In contrast, for CBKP1P2 we measured a Mw of 90 kDa close to the theoretical Mw for a heteropentamer with 1:1:1:1:1 stoichiometry (101 kDa) (Supplementary Table 2).

For the fitting of our SAXS data (Fig. 6), we constructed 3D models of the various complexes starting from high resolution structures of subcomplexes (Kae1-Pcc1, PDB 3ENC; Kae1-Bud32, PDB 3EN9, Bud32-Cgi121, PDB 4WW9) and our present PaPcc1-Pcc2 and PaPcc2-Pcc2 structures. The initial model of the octameric CBKP1[40] resulted in a few slight steric clashes which could be relieved using the FoxsDock software, a method for protein docking against SAXS profiles. The theoretical SAXS curve of the adjusted CBKP1 octamer model fitted very well to the experimental data with a $\chi^2$ value of 1.39 (Supplementary Fig. 6A). The CBKP1 octamer adopts a V-like shape, in line with the experimental maximal extension value of about 188 Å (Supplementary Fig. 6A, middle panel). The Kae1-Pcc1 interface in the model octamer is slightly twisted compared to that in the Kae1-Pcc1 structure (PDB code 3ENO) but we modelled it as the interface of OSGEP-LAGE3 (PDB code 6GWJ).

For modelling the CBKP2 structure, we replaced Pcc1 in the CBKP1 model with PaPcc2 and imposed the Pcc2 homodimer as the central dimerization unit to construct the octamer. This model fitted the experimental SAXS curve satisfactorily with a $\chi^2$ value of 1.64 (Supplementary Fig. 6A, right panel). In stark contrast to the V-shaped CBKP1 model, CBKP2 octamer model adopted an open conformation whereby the two heterotetramers were pointing into opposite directions (Supplementary Fig. 6B). The model of the CBKP1P2 pentamer fits very well the experimental SAXS curve ($\chi^2$ value of 0.619) and is in good agreement with the maximal calculated extension value of 125 Å (Supplementary Fig. 6A, left panel).

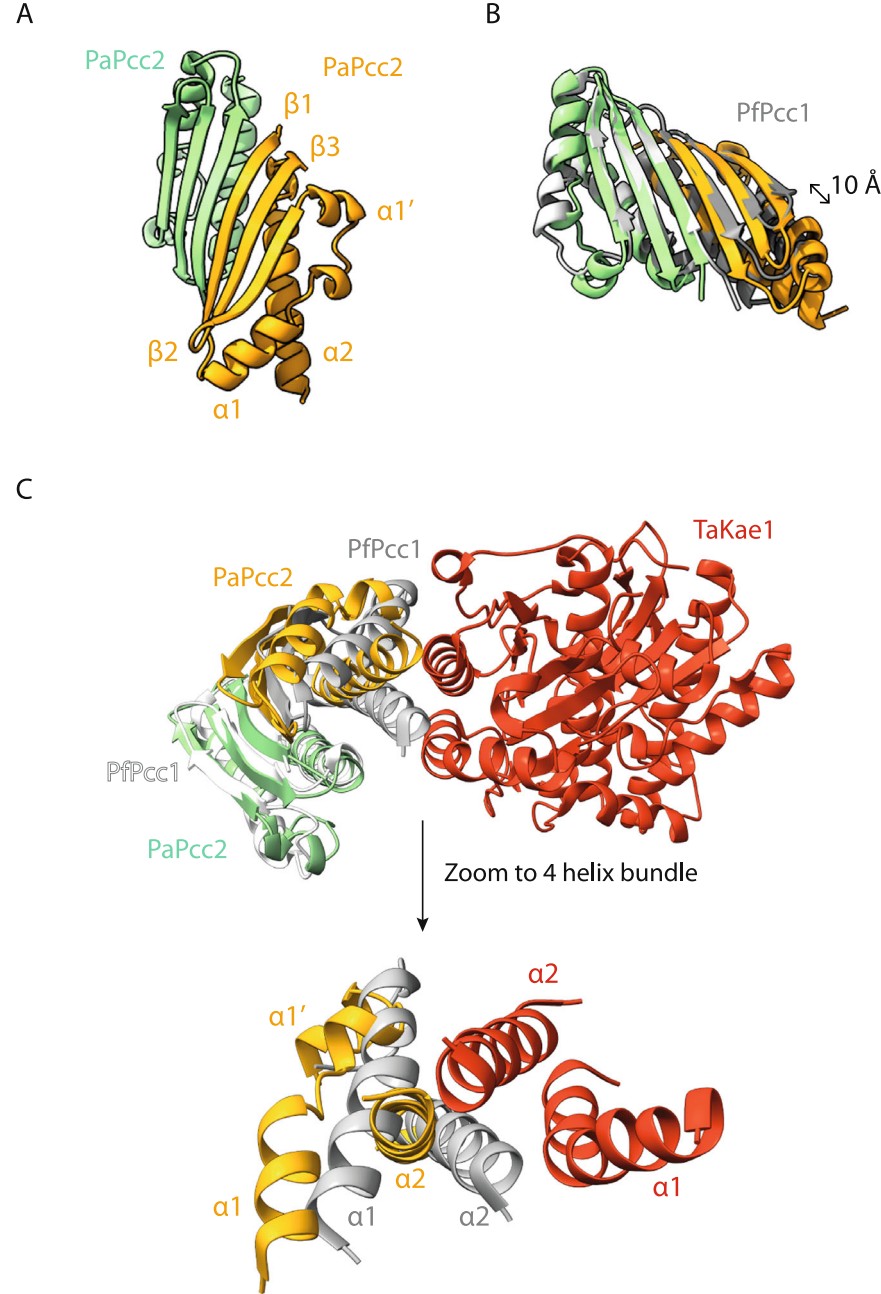

**Fig. 4 | Crystal structure of PaPcc2 homodimer. A** The subunits are in green and yellow, the third copy in the asymmetric unit is not shown. The β1 strands of the green and the orange copies associate in antiparallel to create a continuous β-sheet. The third copy (gray) is loosely associated with the orange copy and this interface is probably purely crystallographic. This copy forms with a crystal symmetry related mate the same type of dimer as the green/orange. **B** Superposition of the PaPcc2 homodimer (green and orange) with PfPcc1 homodimer (grey). The 10 Å longitudinal shift of one PaPcc2 monomer relative to PfPcc1 monomer is indicated with double arrow. **C** Superposition of PaPcc2 homodimer (green and orange) with the crystal structure of PfPcc1 homodimer (grey and white) bound to one copy of TaKae1 (red) from *T. acidophylum* (PDB 3ENO). A zoom to the four-helix bundle composing the PfPcc1-TaKae1 interface is shown below.

Together, the SAXS data confirmed that PaPcc2 prevents the dimerization of CBKP1P2 similar to the effect of Gon7/C14 proteins within the yeast and human KEOPS complexes. Furthermore, the data indicate that in absence of PaPcc1, PaPcc2 induces dimerization of CBKP2 but its quaternary structure is significantly different from that of CBKP1.

## PaPcc2 and PaPcc1 are not isofunctional

We next examined the impact of PaPcc2 on the activity of the pentameric and octameric complexes. To this end, we measured the maximal velocity of t⁶A synthesis and tRNA binding properties for the three complexes.

The kinetic data showed that CBKP1 and CBKP1P2 were active while CBKP2 complex was inactive (Fig. 7A). Notably, we could fully restore the activity of the CBKP2 complex when adding five-fold molar excess of PaPcc1 to the reaction mixture. The reaction could also be partially restored by equimolar concentrations of PaPcc1 suggesting that PaPcc1 has the capacity to displace PaPcc2 from the preformed CBKP2 complex (Supplementary Fig. 7A). For the two active complexes, we compared the maximal reaction velocity as a function of complex concentration. This revealed that the CBKP1 complex exhibited a two-fold higher initial velocity as compared to the CBKP1P2 complex (Supplementary Fig. 7B). Assuming that CBKP1 complex has two catalytic sites per octamer and can process two

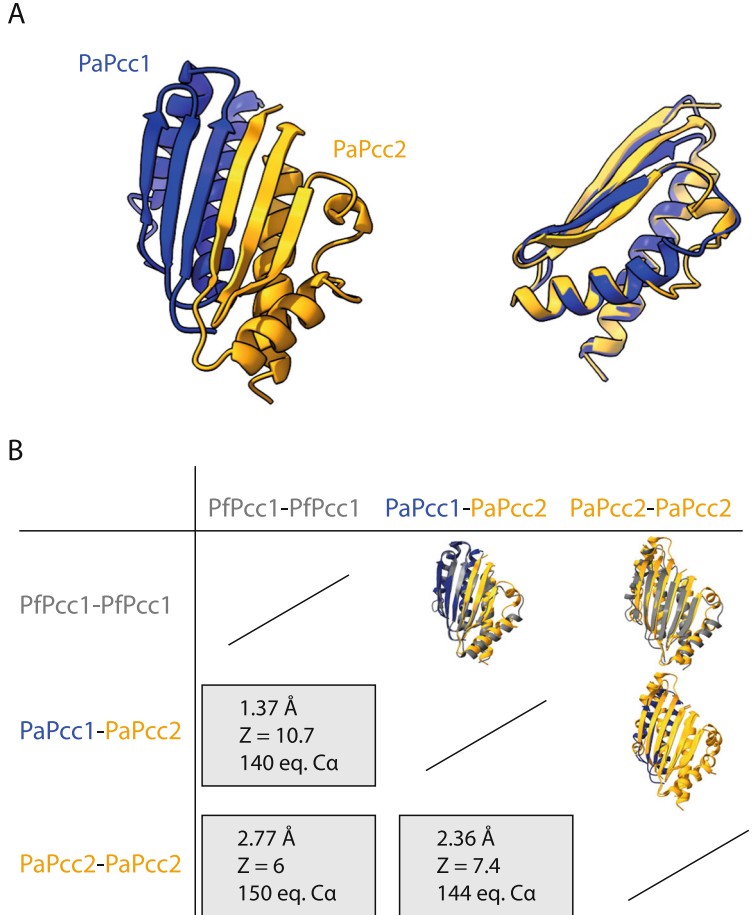

**Fig. 5 | Crystal structure of PaPcc1-PaPcc2 heterodimer. A** Cartoon presentation of PaPcc1-PaPcc2 structure (in blue and orange, respectively) is shown on the left and the superposition of PaPcc2 onto PaPcc1 is on the right. **B** Pairwise comparison of PaPcc1 and PaPcc2 homo and heterodimers. The structures were superposed using ChimeraX 1.4. The RMSD of atomic coordinates and Z score are given for the indicated number of equivalent alpha carbons.

tRNA molecules simultaneously this would indicate that CBKP1P2 and CBKP1 complexes are equally efficient for the catalysis of t⁶A formation.

We next measured the binding affinity of the three complexes toward substrate tRNA using fluorescence anisotropy. We first established that our experimental conditions allowed the binding equilibrium to be achieved (see materials and methods). Titration of the CBKP1, CBKP1P2 and CBKP2 yielded binding equilibrium constants of 13, 100 nM, and ~1800 nM respectively (Fig. 7B, Supplementary Fig. 7C). This suggested that the CBKP2 complex is affected in tRNA binding, which could be one reason explaining why this complex is inactive.

Collectively, the data show that the CBKP1 and CBKP1P2 complexes exhibit robust t⁶A biosynthetic activity and tRNA binding affinities in the nanomolar range. However, CBKP2 has substantially less affinity for tRNA and is inactive indicating that, despite striking structural resemblance, Pcc1 and Pcc2 are not isofunctional. This further suggests that Pcc2 paralogs evolved to provide a specific function as a fifth subunit within the KEOPS complex.

### Search for Pcc2 orthologs in eukaryotes

The evolutionary proximity between archaea and eukaryotes and wide conservation of Pcc2 in archaea raised the possibility that Pcc2 orthologs could also be present in eukaryotes. To test this hypothesis, we performed sensitive sequence similarity searches against eukaryotic protein sequences using the sequence conservation profile of archaeal Pcc2 proteins. This search retrieved only seven eukaryotic sequences passing the significance threshold. However, structure

supported sequence alignment showed that the characteristic pair of acidic residues (D/E⁷², D/E⁷³) was missing in those sequences indicating that these are likely Pcc1 orthologs (Supplementary Fig. 8). The data thus suggested that Pcc2 orthologs are specific for Archaea or that eukaryotic Pcc2 homologs are too divergent to be detected by sequence similarity searches.

To explore a putative evolutionary link between Pcc2 and Gon7/C14, we superposed the 3D structures of the human LAGE3/C14 (LAGE3 is the human ortholog of Pcc1) and the archaeal Pcc1/Pcc2 complexes (Fig. 8A). Despite the fact that Pcc2 and C14 have seemingly a different fold, the similarities of their structures and their interaction with Pcc1/LAGE3 are striking. The two complexes are characterized by a continuous β-sheet covered by long helices that are the main interacting elements. C14 consists of a N-terminal β-hairpin followed by about 40 disordered residues that precede the long α-helix. Pcc2 forms a three stranded sheet with β1β3β2 topology. Superposition of the C14/LAGE3 and Pcc1/Pcc2 complexes perfectly aligns β2 of C14 with β3 from Pcc2. A deletion event in *S. cerevisiae* Gon7 of 39 residues (between position 12 and 51) would yield a Pcc2-like structure. To further explore the possibility that Gon7/C14 proteins could be highly diverged Pcc2 orthologs, we performed structure-guided alignment which showed that Pcc2 sequences can actually be fairly well aligned with full length eukaryotic Gon7 and C14 sequences (Fig. 8B, Supplementary Fig. 9). The majority of the positions with high similarity scores in Pcc1 and Pcc2 sequences are also found in Gon7 and C14 proteins. Notably, Gon7 and C14 proteins contain a D/E⁷³ signature residue (which is never found in Pcc1 proteins) suggesting that they may indeed derive from a Pcc2-like ancestor.

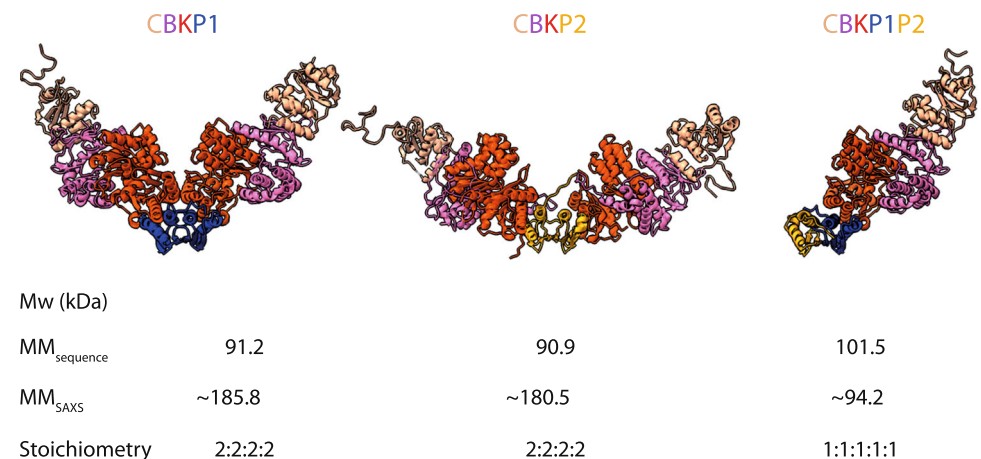

**Fig. 6 | Model structures of PaKEOPS complexes.** The structures of the three complexes modelled against the scattering data are shown. Calculated molecular mass from sequence and from scattering data and the deduced stoichiometry are indicated at the bottom. The subunits are colored as follows: Cgi121 (tan), Bud32 (violet), Kae1 (red), Pcc1 (blue), and Pcc2 (orange) CBKP1 and CBKP2 are dimers of two heterotetramers whereby Pcc1 and Pcc2 serve as dimerization platform, respectively. CBKP2 superdimer adopts much more open conformation with the two heterotetramers pointing into opposite directions. CBKP1P2 is a heteropentamer with Pcc2 protein engaging the Pcc1 via its dimerization interface.

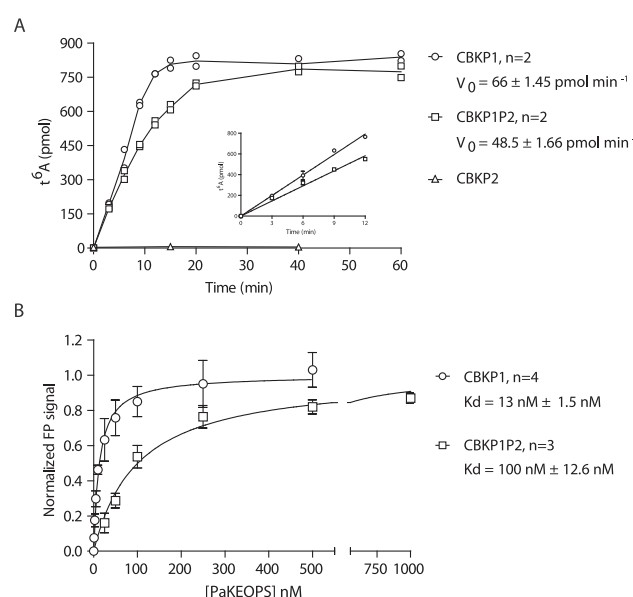

**Fig. 7 | Enzymatic and tRNA binding activity of PaKEOPS complexes. A** The $t^6A$ formation was measured for the three complexes at different time points and the linear part of the curve (inset) was used to calculate the steady-state velocity ($V_0$) of the reaction. All complexes were present at 1 μM concentration. $n = 2$ biologically independent experiments. **B** tRNA binding measurements using fluorescence polarisation. Titration experiments were performed with fixed concentration of fluorescently labelled tRNA$^{Lys}$ (UUU) from *Pyrococcus abyssi* and increasing concentrations of PaKEOPS complexes. The fluorescence signal was normalized using $B_{max}$ values 108.5 and 193.6 for CBKP1P2 and CBKP1, respectively. $n = 4$ (CBKP1) or $n = 3$ (CBKP1P2) biologically independent experiments. Data are presented as mean values +/- standard deviation. Source data are provided as a Source Data file.

If Pcc2 and eukaryotic Gon7/C14 share common ancestry, then the most parsimonious evolutionary scenario postulates that all eukaryotic lineages should be equipped with Pcc2 or Gon7/C14 orthologs. Since we failed to identify Pcc2-like orthologs by sequence similarity searches, we tested if Gon7/C14-like proteins can be found outside opisthokonta (which regroup animals and fungi). Using HMM profile searches derived from representative alignment of Gon7 or C14 orthologs we only identified Gon7 sequences in two closely related

plants *Carpinus fangiana* (hornbeam) and *Quercus suber* (cork oak) belonging to the Fagales order (Supplementary Fig. 10). The phylogenetic analysis placed these sequences within the fungal Gon7 orthologs suggesting that these are not genuine plant Gon7 orthologs but rather genes that were acquired by horizontal transfer from fungi or contamination.

Overall, the data indicate that Pcc2 proteins are specific for Archaea and that Gon7/C14 are either highly divergent orthologs of Pcc2 or a result of convergent evolution whereby opisthokonta had recruited a distinct gene to fulfill a function in regulating the oligomeric state of KEOPS complex.

## Discussion

$t^6A$ biosynthetic pathway is a biological process that appeared before the divergence of life into the three distinct domains. The driving forces behind the evolution of this machinery and the resulting impact on physiology and evolution of cells are not well understood. This is particularly true for the KEOPS complex that evolved in eukaryotes and archaea by acquisition of several accessory proteins to assist the catalytic subunit Kae1. In the present study, we identified an additional accessory protein, which we named Pcc2, as a previously unrecognized core subunit of KEOPS in Archaea.

### Gene duplication is a universal mechanism for the evolution of the $t^6A$ synthetic pathway

We show here that Pcc2 is a paralog of Pcc1 that emerged before the diversification of archaea through an ancient duplication event of the common ancestral gene. This was also concomitantly reported by Wu and colleagues, who, similar to us, noticed a presence of two genes (SiRe_1278 and SiRe_1701) encoding Pcc1-like proteins in the crenarchaeon *Saccharolobus islandicus* REY15A[43]. Occurrence of paralogs within $t^6A$ synthetic machineries was also described in bacteria and eukaryotes. One of the two known accessory subunits in the bacterial system (TsaB) is a truncated version of the catalytic subunit TsaD which lost its catalytic activity[5]. In addition, a few distantly related bacterial lineages harbor YciO, an uncharacterized paralog of the TsaC protein[32,44]. Interestingly, in addition to Pcc1 (dubbed LAGE3), humans encode two additional members of the Pcc1 protein family (CTAG1 and CTAG2) which are expressed in several human tumors and in normal testes and ovaries[22]. It appears therefore that gene duplication followed by mutation is a common mechanism employed by the members of the three domains of life for evolution of the $t^6A$ pathway.

A

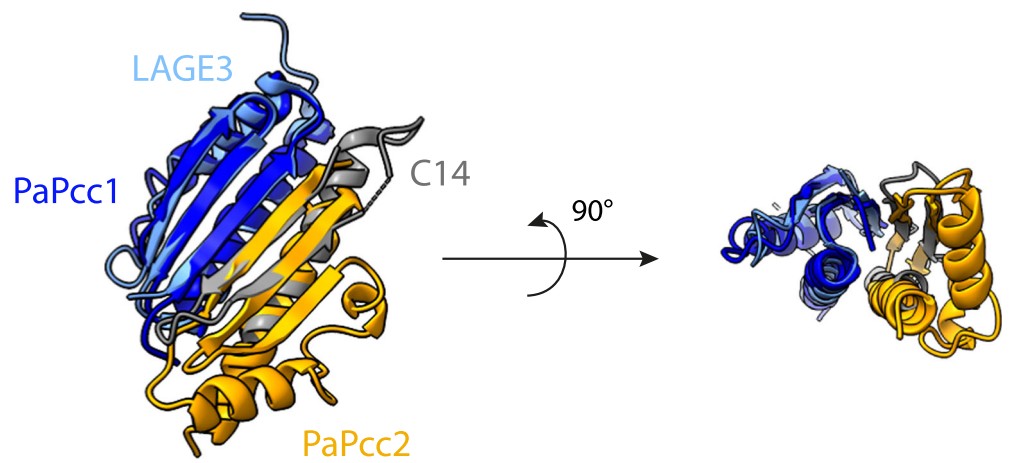

B

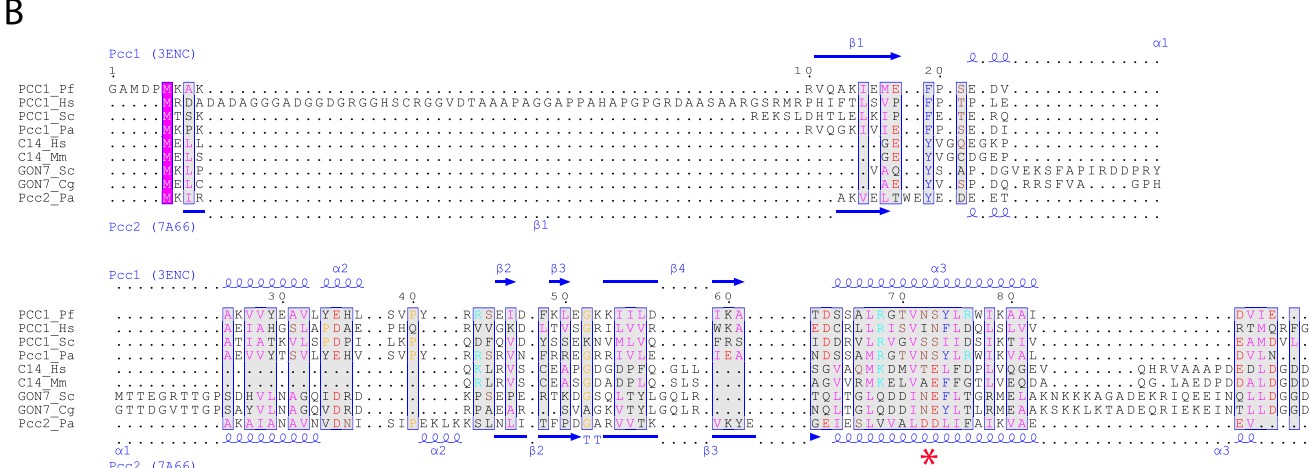

**Fig. 8 | Sequence and structural comparison of Pcc2 and Gon7/C14 proteins.**
**A** Structural comparison of PaPcc1-PaPcc2 and human LAGE3-C14 heterodimers. Superimposition of the crystallographic PaPcc1-PaPcc2 structures (in blue and orange, respectively) on LAGE3-C14 (in light blue and grey, respectively). The left part is a top view showing the continuous 5-strand and 6-strand β antiparallel sheets stabilising the two interfaces in LAGE3-C14 and PaPcc1-PaPcc2 complexes. The right part is a side view showing the α helices that also contribute to the building of the interfaces. **B** Multiple sequence alignment of Pcc1, Pcc2, C14 and Gon7 proteins from representative metazoan (*H. sapiens*, *M. musculus*), fungal (*S. cerevisiae*,

*C. glabrata*) and archaeal (*P. abyssi*, *P. furiosus*) species. Red star denotes the $N^{72}$, S/$T^{73}$ and D/$E^{72}$, D/$E^{73}$ motifs characteristic for Pcc1 and Pcc2 paralogs, respectively. Secondary structure distribution of Pcc1 from *Pyrococcus furiosus* (3ENC) and of Pcc2 from *Pyrococcus abyssi* (7A66) are shown above and below the alignment, respectively. The residues are colored according to their physico-chemical properties. The positions with global similarity score higher that the threshold value of 0.7 are framed in blue. Multiple alignment with an extended set of representative sequences is shown in Supplementary Fig. 9.

## Pcc2 evolved to prevent the formation of KEOPS superdimers

Following the gene duplication event, the two paralogs mainly diverged by accumulating mutations in their C-terminal parts, specifically the α2 helix, which contributes to the Pcc1-Pcc2 interface. We found that the interface statistics (the surface and the number of interactions) of the heterodimer is in favor over Pcc1-Pcc1 or Pcc2-Pcc2 homodimers thus suggesting that Pcc2 evolved to prevent the homodimerization of Pcc1. In line with this, Wu and colleagues reported that adding recombinant Pcc2 from the archaeon *S. islandicus* to the preformed binary Kae1-Pcc1 complex (which has 2:2 stoichiometry) abolished its dimerization[43] indicating, as suggested by us, that the Pcc1-Pcc2 interaction is preferred over the Pcc1-Pcc1 interaction.

Second major difference between Pcc1 and Pcc2 orthologs lies in the α1 helix which in Pcc2 orthologs is split in two perpendicularly oriented smaller helices. This, as well as a significant shift in the position of the α2 helix, suggests a substantially different Pcc2-Kae1 interface compared to the Pcc1-Kae1 interface. Accordingly, the CBKP2

and CBKP1 dimers (as modelled from our SAXS data) adopt two very different conformations. These putative differences at the Pcc2-Kae1 interface could explain why the CBKP1P2 complex does not form a superdimer.

## Pcc2 is essential in some archaea

Using genome-wide transposon mutagenesis Zhang and colleagues identified 441 essential genes in the crenarchaeon *Sulfolobus islandicus* among which figured both Pcc1 and Pcc2[45]. A similar systematic mutagenesis approach in an euryarchaeon *Methanococcus maripaludis* found however that Pcc1 was essential but not Pcc2[46]. A targeted genetic investigation of KEOPS complex in another euryarchaeal model organism *Haloferax volcanii* (which also encodes Pcc2, HVO_1146), reported that the Pcc1-encoding gene (HVO_0652) could not be deleted using markerless pop-in pop out strategy, and the mutants obtained by marker replacement were very slow-growing[17]. These in vivo data indicate, in line with our findings, that Pcc2 cannot functionally complement Pcc1 and therefore has

evolved to provide a distinct function. Moreover, this function is, at least in some archaea, essential most likely because it is required for t⁶A synthesis in vivo. Whether Pcc2 associates with the KEOPS complex in archaea in vivo was not addressed in this work, however, this was investigated by Wu and colleagues in *S. islandicus*. The authors expressed the His-tagged versions of Pcc1, Kae1 or Bud32 in *S. islandicus* and performed affinity purification from cell lysates coupled with mass spectrometry. In the three assays Kae1, Pcc1, and Pcc2 co-purified suggesting that the five subunit complex is the natural version of KEOPS in archaea[43].

The enzymatic assays conducted in this work show that Pcc2 does not potentiate the t⁶A activity in vitro suggesting that it has no direct influence upon the catalysis. The Sicheri group showed that the C14 subunit increased the t⁶A activity of the human KEOPS complex by 3-4 fold in vitro but it was suggested that this may be attributed to an indirect effect through stabilization of the complex[37]. Indeed, the experiments in human cell lines established that mutations in C14-encoding gene (but not in Kae1 ortholog OSGEP) lead to the decrease in protein levels of the other four KEOPS subunits[28]. Incidentally, we noticed during the purification of the KEOPS complexes that the purified pentameric complex is more stable in solution than the octameric CBKP1 complex which has a tendency to form aggregates. Pcc2 may therefore indirectly impact the intracellular t⁶A levels by stabilizing the KEOPS complex.

### Pcc2 and Gon7/C14 proteins may be homologs

Several observations suggest that Pcc2 and eukaryotic Gon7/C14 may be homologs. Sequence alignments show that Pcc1 and Pcc2 stretches with high sequence similarity scores can be almost systematically aligned with Gon7/C14 sequences. This is particularly true for the α2 helix of Pcc1/Pcc2 which (together with β1 strand) provides the main dimerization interface. The Gon7/C14 proteins harbor a Pcc2-type signature residue D/E[73] which is never present in Pcc1 sequences pointing to a possible common ancestry with the Pcc2 proteins. Consistent with this hypothesis, structural alignment of Pcc1/Pcc2 with human LAGE3/C14 complex shows a remarkable conservation of the interaction interface despite the fact that the β-sheets of C14/Gon7 and Pcc2 proteins have a different topology (β1β3β2 for Pcc2 and β1β2 for C14/Gon7) suggesting that this interface is a functionally important requirement that has been conserved from archaea to eukaryotes. Beyond sequence and structural similarities, Pcc2 has a shared function with eukaryotic Gon7/C14 in preventing the formation of KEOPS superdimers.

## Concluding remarks

Understanding the purpose behind the recruitment of several accessory proteins to support the Kae1 catalytic function is one of the most exciting and still poorly understood mysteries in the KEOPS field. The discovery of Pcc2 now suggests that the 5-subunit KEOPS is the ancestral form of the complex that emerged before the diversification of archaea and was vertically transmitted to eukaryotes from an archaeal ancestor. The possibility to prevent the dimerization of the KEOPS complex thus seems to be a common principle in both archaea and eukaryotes. It is therefore plausible to think that highly divergent (in sequence but not in structure) orthologs of Gon7/C14 exist in many if not all eukaryotic lineages. Understanding how Pcc2 regulates the KEOPS function in vivo in archaea is one of the exciting future prospects of this work with possible implications for the eukaryotic system.

## Methods
### Cloning procedures

The genes encoding KEOPS complex of the archaeon *Pyrococcus abyssi* (PaKEOPS) PAB_RS09585 (PaKae1), PAB_RS08575 (PaBud32), PAB_RS06930 (PaCgi121) and PAB_RS01470 (PaPcc1) were amplified by polymerase chain reaction (PCR) using pKEOPS500 plasmid (Perrochia et al., 2013a) as DNA template. The gene PAB_RS00380 encoding PaPcc2 was amplified by polymerase chain reaction (PCR) using genomic DNA of *P. abyssi* GE5 strain as template. Different combinations of bicistronic or polycistronic sequences were cloned into pET26b (Novagen) via *Nde*I-*Xho*I restriction sites either using T4 ligase (NEB) or by Gibson assembly (NEB) following the protocol supplied by the manufacturer.

pKP1His6 and pKP2His6 contain each a bicistronic sequence with PaKae1 plus PaPcc1 or PaKae1 plus PaPcc2 encoding genes, respectively. The genes encoding PaPcc1 and PaPcc2 are fused at their 3′ end to a sequence encoding a hexa-histidine tag.

pKP2Strep-P1His6 contains a tricistronic sequence with PaKae1, PaPcc2 and PaPcc1 encoding genes. The gene encoding PaPcc2 is fused at its 3′ end to a sequence encoding a Strep-tag whereas the gene encoding PaPcc1 is fused at 3′ end to a sequence encoding a hexa-histidine tag.

pCBK and pCB contain PaKae1, PaBud32 and PaCgi121 encoding genes or PaBud32 and PaCgi121 encoding genes respectively. These genes are not fused to an epitope tag encoding sequence.

### Heterologous expression and purification of recombinant proteins

Recombinant proteins were expressed in *E. coli* Rosetta2 (DE3) pLysS strain (Novagen). The cells were transformed either with pCBK, pCB, pKP1His6, pKP2His6, or with pKP2strepP1 His6 plasmids and were grown at 37 °C in LB medium with kanamycin (50 μg/ml) and chloramphenicol (25 μg/ml) until the optical density (600 nm) reached 0.6–0.7. Gene expression was induced by addition of IPTG (1 mM), and incubation was continued at 37 °C for 2 hours. Cells were collected by centrifugation (5000 *g*, 15 min) and resuspended in cold lysis buffer (20 mM Tris-HCl pH 8, 500 mM NaCl, 5 mM ß-mercaptoethanol, 10% Glycerol) supplemented with complete EDTA-free protease inhibitors cocktail (Roche). Cell pellets containing different subcomplexes were mixed in defined ratio (see below) and lysed at 4 °C with a one-shot cell disruptor (Constant Systems). The lysate was clarified by centrifugation at 30,000 g for 30 min at 18 °C. The clarified lysate was incubated 10 to 15 min at 65 °C to denature the majority of *E. coli* proteins. The precipitates were removed by centrifugation at 30,000 g for 20 min at 18 °C. The soluble fraction was supplemented with 10 mM imidazole and His-tagged proteins were purified by affinity chromatography on NiNTA column (Qiagen) at room temperature. Equilibration, washes, and elution steps were done with lysis buffer containing respectively 10, 40, and 400 mM imidazole. Fractions of interest were pooled, concentrated at 18 °C and submitted to size exclusion chromatography on a HiLoadR 16/600 SuperdexR 200 pg or a HiLoadR 16/600 SuperdexR 75 pg (GE Healthcare) equilibrated with 20 mM Tris-HCl pH 8, 500 mM KCl, 10% Glycerol and 5 mM ß Mercaptoethanol.

In order to reconstitute and purify different PaKEOPS complexes, cells transformed with different recombinant plasmids (see cloning procedures) were mixed in defined cell pellet mass ratio and lysed together.

PaPcc1-PaPcc2 and PaPcc2 proteins used for SAXS analysis and crystallization were purified from pCBK transformed cells mixed with pKP1P2 transformed cells (cell mass ratio 2:1) and from pCBK transformed cells mixed with pKP2 transformed cells (cell mass ratio 2:1), respectively. In these conditions, significant excess of PaPcc1-PaPcc2 complex and PaPcc2 homodimer over the complete PaKEOPS complexes is obtained when applying the two-step purification protocol described above.

The PaKEOPS complexes used for SAXS analysis and enzymatic assays were purified from pCBK transformed cells mixed with pCB transformed cells and either pKP1His6 transformed cells (cell mass ratio 2:1:1) or pKP2His6 transformed cells (cell mass ratio 2:1:1) or pKP1His6P2Strep- transformed cells (cell mass ratio 2:1:1).

**Sequence similarity searches for identification of Pcc1 and Pcc2**
To search for Pcc1 and Pcc2 orthologs, we performed iterative searches against UniProtKB database using the Jackhmmr program (https://www.ebi.ac.uk/Tools/hmmer/search/jackhmmer) with default settings. We manually selected 21 representative archaeal species each encoding both Pcc1 and Pcc2. We then generated separate alignments for Pcc1 or Pcc2 sequences using MAFFT[47] and used those as input. After two iterations, we retrieved 1264 significant hits that could be unambiguously assigned as archaeal Pcc1 (646 proteins) or Pcc2 (618 proteins) based on e-values. The table containing the raw data and the summary of the search results is available as supplementary file.

To generate the sequence logos specific for Pcc1 or Pcc2 proteins we chose from the above dataset 27 sequences representative of the archaeal diversity. When possible, we chose species with complete genomes, otherwise, metagenome-assembled genomes (MAGs) were chosen for which the Pcc1 and Pcc2 encoding genes were found within the same MAG. The sequences are available on the figshare website under https://doi.org/10.6084/m9.figshare.21640394.v1

**Phylogenetic analysis**
For the phylogenetic analysis, we chose couples of Pcc2 and Pcc1 paralogs from one representative of 27 phyla of Archaea covering the entire archaeal diversity. Protein alignment was done with MAFFT V7 with the amino acid matrix Blosum 30 and iterative refinement methods L-INS-I or G-INS-I[47]. The trimming was performed with BMGE with a BLOSUM30 matrix and the -b 1 parameter[48]. The Maximum likelihood trees were constructed using the IQ-TREE v1.7 software (http://www.iqtree.org/)[49] with the best substitution model as suggested by the ModelFinder option -MFP. The branch robustness was estimated by the SH-aLRT approximate likelihood ratio test[50], the ultrafast bootstrap approximation[51] (10000 replicates for each), and with the transfer bootstrap expectation approaches[52] based on 500 boot trees.

Sequences of Gon7 orthologs from Metazoa (ENOG503BUEY, 60 sequences) and Fungi (ENOG503P560, 32 sequences) were retrieved from EggNOG database 5.0.0[53] which harbours sequences selected for diversity and filtered by genome quality. Sequences were aligned with the T-coffee webserver[54] and the tree was inferred using IQ-TREE v1.7 software using automatic selection of sequence evolution model.

**Crystallization of the PaPcc1-PaPcc2 heterodimer**
PaPcc1-PaPcc2 complex was purified by affinity chromatography on NiNTA column (Qiagen) at room temperature followed by size exclusion chromatography as described above except that the lysis buffer contained 300 mM NaCl. Fractions containing the heterodimeric PaPcc1-PaPcc2 complex were pooled and concentrated to 7.9 mg mL$^{-1}$ for crystallization trials. Crystals were obtained by mixing 100 nL of the protein solution with 100 nL of 15% PEG 6K and 5% glycerol using the sitting-drop vapor diffusion method. The crystallization drop was equilibrated against a 70 µL reservoir solution of 15% PEG 6K and 5% glycerol. Drops were incubated at 18 °C and crystals appeared after about 21 days. Crystals were cryo-protected by quick-soaking in 20% PEG 6 K and 20% glycerol prior to flash freezing in liquid nitrogen.

**Crystallization of the PaPcc2 homodimer**
PaPcc2 homodimer was purified using the same procedure as for PaPcc1-PaPcc2 heterodimer. Fractions containing pure PaPcc2 were concentrated to 22 mg.mL$^{-1}$ for crystallization trials. Crystals of PaPcc2 were obtained by mixing 100 nL of the protein solution with 100 nL of 30% PEG 3 K 0.2 M lithium sulfate and 0.1 M Tris HCl pH 8.5 using the sitting drop vapor diffusion (crystals appeared after 17 days of incubation at 18 °C). The crystallization drop was equilibrated against a 70 µL reservoir solution of same the precipitant. Crystals were cryo-protected by quick-soaking in 50% paraffin and 50% paratone prior to flash freezing in liquid nitrogen.

**Structure solution and refinement**
X-ray diffraction data collection was carried out at Synchrotron SOLEIL on beamline Proxima 2 (Saint Aubin, France) at 100 K. Data were processed, integrated and scaled with the XDS software package[55]. The PaPcc2 crystals belong to space group C12$_1$ with unit cell a = 76.80 Å b = 81.31 Å c = 66.48 Å and α = 90° β = 121.59° γ = 90°. The structure of PaPcc2 was solved by molecular replacement using PHASER[56] implemented in the CCP4 suite. A search model for PaPcc2 was obtained by MODELLER[57] using the known structure of PfPcc1 (PDB 3ENC). 3 copies of PaPcc1 were found in the asymmetric unit of the crystal. The initial structure was refined with BUSTER and interactively manually adjusted with COOT[58]. The PaPcc1-PaPcc2 crystals belong to space group P3$_2$ 21 with parameters of unit cell of a = 69.898 Å b = 69.898 Å c = 92.751 Å and α = 90° β = 90° γ = 120°. One copy of the heterodimer was found in the asymmetric unit. The initial structures were refined with BUSTER and interactively manually adjusted with COOT[58]. Data reduction and refinement statistics are gathered in the Supplementary Table 1.

**SEC-SAXS analysis**
SAXS experiments were carried out on the SWING beamline at SOLEIL synchrotron (Saint-Aubin, France). The sample to detector (Eiger 4 M Dectris) distance was set to 1500 mm, allowing reliable data collection over the momentum transfer range 0.005 Å$^{-1}$ < q < 0.5 Å$^{-1}$ with q = 4πsin θ/λ, where 2θ is the scattering angle and λ is the wavelength of the X-rays (λ = 1.0 Å). To collect data on homogenous protein samples, SAXS data were collected at the exit of a size exclusion high-performance liquid chromatography (SEC HPLC Bio-SEC3 Agilent) column directly connected to the SAXS measuring cell. 65 µL of PaCgi121/PaBud32/PaKae1/PaPcc1/PaPcc2 (CBKP1P2), CBKP1, and CBKP2 samples concentrated at 2.55, 0.72, and 3 mg/ml, respectively, were loaded into the column equilibrated with 20 mM Tris pH 8.5, 300 mM NaCl, and 5 mM 2-mercaptoethanol and 5% glycerol. Flow rate was set at 300 µL/min, frame duration was 1.0 s, and the dead time between frames was 0.01 s. The protein concentration was estimated by UV absorption measurement at 280 and 295 nm using a spectrophotometer located immediately upstream of the SAXS measuring cell. A large number of frames were collected before the void volume and averaged to account for buffer scattering. SAXS data were normalized to the intensity of the incident beam and background (i.e., the running buffer) was subtracted using the program FoxTrot50, the Swing in-house software. The scattered intensities were displayed on an absolute scale using the scattering by water. Identical frames under the main elution peak were selected and averaged for further analysis. Radii of gyration, maximum particle dimensions, and molecular masses were determined using PrimusQT[59].

**Modelling of the complete KEOPS complex**
The PaKEOPS complexes were modelled using a combination of structures of archaeal KEOPS subcomplexes and of the present PaPcc1-PaPcc2 structure by the Modeller software[60]. The Dadimodo[61] and FoxsDock[62] programs were then used to adjust the models of CBKP1P2, CBKP1, and CBKP2 to optimally fit the experimental curve I(q). The final adjustment was performed using the program CRYSOL[63].

**Comparative sequence and structure analysis**
Multiple alignment of the Pcc1 and Pcc2 sequences was built using the Blosum62 matrix with MAFFT vs. 7[47] (Katoh et al., 2019). The sequence conservation was rendered using WebLogo 3 (http://weblogo.threeplusone.com/)[64]. Structure-based alignment of a subset of these members was performed using the ESPRIT web-server (https://espript.ibcp.fr/ESPript/ESPript/)[65]. All figures of structures were generated with Pymol (The PyMOL Molecular Graphics System, Version

2.0 Schrödinger, LLC). Structure similarity comparisons were performed with the PDBeFold web server (https://www.ebi.ac.uk/msd-srv/ssm/). Surface of protein-protein interface and the number of potential hydrogen bonds was calculated using PDBePISA web server (https://www.ebi.ac.uk/pdbe/pisa/).

## tRNA production and purification

tRNA$^{Lys}$ (UUU) from *P. abyssi* was produced by run-off transcription from a pUC18 vector containing a tRNA-encoding gene under the control of the T7 promotor. The transcription unit (TU) consists from 5′ to 3′ end of: hammerhead ribozyme carrying T7 promotor, tRNA gene, glmS ribozyme and a double MS2 tag[66]. The complete sequence of the TU is shown in Supplementary Fig. 11. This construct allows to produce, after transcription and ribozyme cleavage, tRNA molecules with precisely defined 5′ and 3′ extremities. The transcription reaction was carried out in the final volume of 10 mL and it contained 0.02 μg of *Bpi*I-linearized template plasmid, 40 mM Tris-HCl pH 8, 1 mM spermidine, 0,01%, Triton X-100, 5 mM DTT, 2 mM of each NTP, 10 mM of MgCl$_2$, 80 U of RNasine (Promega) and 0.03 μg/μL of T7 RNA polymerase. The mixture was incubated for at least 4 hours at 37 °C and ribozymes were removed by addition of glucosamine-6-phosphate (1 mM final concentration) and MgCl$_2$ solution (5 mM final concentration) and incubation for 20 min at room temperature. The reaction was stopped by heating at 80 °C for 10 min. The reaction mixture was concentrated by ultrafiltration (10 kDa cut-off filter) to a final volume of 1 mL and the tRNA was purified on a 10% acrylamide gel (19:1) containing 8 M urea. The band containing tRNA was excised and the tRNA was extracted by crush and soak method with 5 mL of extraction solution (0.3 M sodium acetate pH 5.3, 0.1% SDS (w/v), 1 mM EDTA pH 8.0). The tRNA-containing solution was concentrated by ultrafiltration using 3 kDa cut-off filter to about 500 μL and precipitated by addition of two volumes of cold absolute ethanol. After centrifugation, the tRNA pellet was washed twice with 70% ethanol (v/v), resuspended in water, and stored at −80 °C.

## In vitro assay for the synthesis of t⁶A-modified tRNA

The assays were performed as previously described[30] with minor modifications: briefly, the reaction mixture (final volume of 25 μL) contained 70 μM C$^{14}$ L-threonine (55 mCi/mmol, Isobio), 100 μM MnCl$_2$, 5 mM Na$_2$CO$_3$ and 10 μM of Pa_tRNA$^{Lys}$ (UUU) in 50 mM Tris-HCl pH 8, 200 mM KCl, 5 mM MgCl$_2$, 5 mM DTT. PaKEOPS complexes and PaSua5 were typically added to 2 μM and 0,5 μM final concentration, respectively, unless stated otherwise. For the initial velocity measurements, the CBKP1 and CBKP1P2 complexes were added at 1 μM final concentration. The reaction mixture was incubated for indicated time at 55 °C and the reaction was stopped by addition of 1 mL of 15% (w/v) of trichloroacetic acid (TCA) and incubated on ice for 1 h. Precipitated material was retained on pre-wet glass microfibre GF/F filters (Whatmann) using vacuum apparatus (Millipore). Filters were washed with 2 mL of 5% TCA and 3 mL of 95% EtOH and dried. Liquid scintillation counting was used to quantify the amount of t⁶A-modified tRNA using the standard curve to convert CPM to pmol of tRNA (1 pmol of tRNA = 97.8 CPM).

## Fluorescence anisotropy measurements

Fluorescence anisotropy was used to determine the equilibrium binding constant for tRNA-PaKEOPS complexes. The binding mixture contained, in a final volume of 100 μL, 20 mM Tris, HCl pH 8, 150 mM KCl, 3 mM MgCl$_2$, 1% glycerol, increasing concentration of PaKEOPS complexes, and 1 nM of Pa_tRNA$^{Lys}$ (UUU) labelled at its 3′ end with fluorescein (Horizon). We measured the polarisation signal between 1 and 50 min at 20 °C and concluded that the binding equilibrium was reached within 1 minute. For the binding assays, samples were incubated at 20 °C for 15 min and the fluorescence polarisation was

measured in 96-well half area flat bottom non-treated black polystyrene microplates (Costar) using Infinite M1000 Pro (Tecan) microplate reader set to 470 nm excitation wavelength and 530 nm emission wavelength. All measurements were done at least in triplicates. After subtracting the polarization values obtained for Pa_tRNA$^{Lys}$ alone, fluorescence polarisation in arbitrary units was plotted against protein concentration and the resulting saturation curve was fitted to one-site specific binding equation (GraphPad Prism) Y = Bmax*X/(Kd + X), where Y is the difference between the anisotropy of bound and free tRNA, Bmax is the maximal polarisation signal, X is KEOPS concentration and Kd is the equilibrium dissociation constant.

## Reporting summary

Further information on research design is available in the Nature Portfolio Reporting Summary linked to this article.

## Data availability

The manually curated set of Pcc1 and Pcc2 sequences used in this study are available at https://doi.org/10.6084/m9.figshare.21640394.v1. The raw data from HMM-based searches are available as Supplementary Data 1. The crystal structures of Pcc1 and Pcc2 proteins were deposited to the PDB database, accession numbers 7A66 and 7A67. Source data are provided with this paper.

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

## Acknowledgements

We acknowledge the support of the SOLEIL (St Aubin) and ESRF (Grenoble) synchrotron beamline staffs. We thank S. Planquel of the I2BC crystallization platform for technical assistance. This work was funded by Agence Nationale de la Recherche, grant number ANR-18-CE11-0018 to T.B. and by the French Infrastructure for Integrated Structural Biology (FRISBI) (ANR-10-INSB-05–01) to H.v.T.

## Author contributions

S.M., B.C., and H.v.T. designed SAXS and protein crystallization assays. S.M. performed SAXS and protein crystallization assays and solved protein crystal structures. N.L. provided technical help. V.D.C. and T.B. designed and performed bioinformatic and phylogenetic analyses. M-C. D. and T.B. designed protein purification procedures, enzymatic activity assays, and fluorescence anisotropy assays. M-C. D. performed protein purification, enzymatic activity assays, and fluorescence anisotropy assays. M-C. D., S.M., V.D.C., B.C., H.v.T., and T.B. wrote the manuscript.

## Competing interests

The authors declare no competing interests.
