## [Peer Review File · Nature Communications]

A paralog of Pcc1 is the fifth core subunit of the KEOPS tRNA-modifying complex in ArchaeaReviewer #1 (Remarks to the Author):

The manuscript "A paralog of Pcc1 is the fifth core subunit of KEOPS complex in Archaea" describes first in depth biochemical and structural characterization of an additional Pcc1-like subunit of essential in all three domains of life tRNA modification multisubunit enzyme called KEOPS. This is a very thorough, high-quality work. All key conclusions are strongly supported, and the text is clearly written. I also find convincing the presented evidence that the fifth eukaryotic subunit of KEOPS complex GON1 is a very diverged homolog of the Pcc1-like protein (or Pcc2 as it was named by authors), despite absence of easily detectable sequence and structural similarity. I have therefore only minimal criticism concerning some computational analyses and description of some computational methods.

Minor issues for correction/modification

1. Please provide details for the procedure of discrimination between Pcc1 and Pcc2 families and additional evidence supporting data shown in the Figure 2A. There are three options 1) show a phylogenetic tree with two respective branches indicated and protein ID and species name indicated for all leaves; 2) Alternatively, a complete multiple alignment with differentiating box2 shown can be provided; 3) For all proteins scores for two Pcc1 and Pcc2 HMM profiles and desirably a signature/sequence for box 2 can be provided in a Supplementary Table. For phylogenetic analysis it is OK to make non-redundant set using some objective criteria (eg. one representative for a cluster of 90% identical proteins) and remove fragments, but otherwise all proteins from the two families identified in all archaeal genomes should be included. In particular I am concerned by results for Crenarchaea where Pcc1 subunit is found in much smaller number of genomes than Pcc2. If this is indeed so, at least a note or, better, an explanation/discussion in the text should be included. Also please indicate an overall number of archaeal genomes and specify the database used to make the analysis shown in this figure. And provide HMM profiles or respective multiple alignments if they indeed discriminate these two families well.
2. Indicate how those 27 representatives were selected in either Methods section or all relevant Figure legends. Were they chosen manually?
3. The section on GON1 homologs in plants is beyond the scope of this work, I think. But if it is to stay, contamination of plant genomes by fungi contigs should be ruled out, because, unfortunately, this is the null hypothesis for this observation but not horizontal transfer from fungi to plants. To rule this out taxonomic affiliation of best hits for a dozen of neighboring proteins encoded up- and downstream from respective genes in two identified "plant" contigs should be shown in the supplement. If results will be inconclusive, just include "or contamination" in the sentence in line 653.
4. Resolution and/or font of several supplementary figures is very low (eg. SF2, SF3, SF9). Please fix this.

Minor and optional suggestions:

1. Pcc1 and Pcc2 are similar enough to be annotated in most databases as Pcc1. So, this is a trivial finding and does not require too much space to explain. It is just enough to say that most of archaeal genomes have two Pcc1-like proteins, but function is known only for one of them. Thus, consider streamlining this section of the text and removing Supplementary figure 1.
2. Include a reference for the statement in line 100.
3. Modify "small uncharacterized protein that showed significant sequence similarity" to "functionally uncharacterized homolog" (line 128), because technically most of them are annotated as Pcc1 in protein databases.
4. Line 276. "was realized" correct to "was built"
5. Line 613. Please explain what "seemingly a different fold" means. A topology diagram would help here.
6. Line 673. Delete ",," after TsaB.

Reviewer #2 (Remarks to the Author):

**A paralog of Pcc1 is the fifth core subunit of KEOPS complex in Archaea
Daugeron Marie-Claire^{1,§}, Missouri Sophia^{1,§,§}, Da Cunha Violette^{1,£}, Lazar
Noureddine¹,
Collinet Bruno^{1,2} van Tilbeurgh Herman^{1,#} & Tamara Basta¹**

The authors describe the assignment of a 5th subunit in the archaeal form of the tRNA modifying enzyme complex KEOPS. They make a good case of the evolutionary relations between Archaea and Eukaryotes. The data they present is sound and convincing. They use the KEOPS genes from *P. abyssi* recombinantly expressed in *E. coli* as their model system. They clearly show that all 5 subunits form a functional complex and that the PCC1 and Pcc2 are responsible for the oligomeric states of the enzyme. The authors make a very interesting case that the 5 subunit KEOPS is an ancestral system, maybe part of the LUCA? I particularly like the implication this has for the ever expanding groups of Archaea.

Although I think the research is interesting and sound I have 10 major points to make. These need to be addressed in order to move forward in *Nature Comm* or other journal.

- 1-the title contains two abbreviations which I think is for a broad audience unacceptable. The tRNA modification function needs to be present in the title.
- 2-the introduction does not give a quick overview of the bacterial system which does not utilize a 5 subunit complex. Please give a short description on the bacterial system to be used to discuss later in the evolutionary path of KEOPS.
- 3-in the legend of figure 1 B it is not clear what is meant with the oligomeric state (also in the introduction). Where and when do you have physiologically active dimeric or monomeric complexes in vivo eg monomeric only active eukaryotes, Archaea both monomeric and dimeric, it gets confusing quickly?
- 4-the authors use a short 80-90 residue protein for phylogenetic analysis, which I think is sound, but do not mention the difficulty which such short sequences brings in phylogenetic analysis.
- 5-Figure 2. Panel C is meaningless move to the supplement and please annotate a little bit. I like panel B but extend the structural top bit to the whole sequence and show the zoom in with the sequence logo. Move panel A to the supplement figure 2, I like supplement figure 2 much better but make it a little prettier, better to read and for god sake where are the Crenarchaeota?
- 6-the authors really need to address the variation in the Archaea some have only Pcc1 some have 1 and 2 and some have neither, how does this work do they still have a KEOPS or do they have the bacterial system? Which becomes important in the evolutionary path in the discussion
- 7-Figure 3 they authors show a chromatogram trace of the gel filtration column, almost like copied and past out of the software that runs the equipment. That is just lazy and unacceptable, make a nice figure and put in the appropriate sizes obtained with the gel filtration analysis.
- 8-I got very confused by the SAXS data, how is correlated to the gel filtration data, is it the same or different what is obtained from the SAXS data and gel filtration, any problems with not homogenous samples for SAXS? A table with the different oligomeric states in the different complexes will be helpful.
- 9-Note added in proof? What proof? this is the first version of the manuscript. I point my attention to a manuscript in a preprint server you will have to discuss it. You cannot just mention it as a quick note that is unacceptable (I do not like preprint servers either but you have no choice in that). Please address this preprint manuscript properly as it significantly overlaps with this work and might even strengthen it.
- 10-The authors have not convinced me of the necessity of dimerization prevention in Archaea (Eukaryotes are clearer in that) as described in the conclusion. This is especially confusing as not all Archaea seem to have Pcc2 please enlighten the reader on that.

Reviewer #3 (Remarks to the Author):

The study by Daugeron and coworkers describes the identification and biochemical and structural characterization of a novel archaeal auxiliary protein factor (Pcc2) that is part of the KEOPS complex. The KEOPS complex catalyzes the second step of the t6A modification at position 37 of most tRNAs with UNN anticodons, an essential and universal post-transcriptional modification. The archaeal KEOPS complex was thought to consist of 4 distinct protein components (Kae1, Bud32, Pcc1, and Cgi121), contrasting with the eukaryotic complex that contains 5. In this work, the authors used co-purification and biophysical (X-ray crystallography and anisotropy) approaches to show that Pcc2, a paralog of Pcc1, may bind to the KEOPS complex by directly interacting with either the Pcc1 or the Kae1 subunit (the catalytic site). The crystal structures of Pcc2 and the Pcc2-Pcc1 complex were solved, providing detailed insights into the interactions of these factors. SAXS-based structural models of the KEOPS complex in the presence and absence of Pcc2 or Pcc1 suggest that Pcc2 prevents the dimerization of the KEOPS complex and illustrate how it may function. While the presence of Pcc2 decreases the tRNA binding affinity, it does not significantly affect the activity of the KEOPS complex. Notably, the absence of Pcc2 does not impact the KEOPS complex's activity.

Overall, this work could represent an exciting discovery of interest for the field that may be of interest to a broad readership. The manuscript is well-written, and the data generally support the conclusions. However, there are a few significant gaps in the study regarding the biological aspects of the work, which detracts from its significance. A major gap is that Pcc2 was shown to bind to the KEOPS complex when over-expressed heterologously in *E. coli*; it is unclear whether this is a bona fide interaction in the host. This is important because Pcc2 may not be an exclusive component of the KEOPS complex as it may have alternative functions and interacting partners. This could reconcile the fact that Pcc2 is not necessary for the t6A modification *in vitro* but is essential for cell viability, as discussed by the authors. Thus, this interaction should be validated in the host species. Another critical gap is that there needs to be a robust discussion about the biological function of Pcc2. The authors pose that Pcc2 evolved to prevent dimerization of the KEOPS complex, but no explanation of the biological relevance is provided. While the biochemical and structural data presented are interesting and provide mechanistic insights into how the complex may operate, information on the biological importance of Pcc2 is needed, which would substantially strengthen the significance of this work. Additional recommendations to improve this study are outlined below.

- Important controls are missing in the co-purification experiments to prove that the binding of Pcc2 is specific. For example, can Pcc2 co-purify with a complex missing Kae1?
- Based on their phylogenetic analyses, the authors propose that Pcc2 and Gon7 are functionally related homologs. This hypothesis could be tested using their established co-purification binding assays. Can Gon7/C14 replace Pcc2?
- Can the authors comment on the potential biological role of Pcc2? Is it a regulatory factor? Why is it important for the Pcc2 to prevent KEOPS superdimers? Is it co-expressed with the other KEOPS factors in the host organism? Could it play a role in the first step of the modification reaction?
- Overall, the methodology section provides enough details to reproduce the experiments. However, the sequence of the DNA template (including the ribozyme) for the *in vitro* transcription of the tRNA substrate should be included in the manuscript.
- Can the authors comment on the evolutionary implications of the absence of Pcc2 in

some species?

We would like to thank the three reviewers for their time and investment in providing a thorough assessment of our work and the constructive criticism. Below are point per point answers to reviewer's comments. We feel that the revised version of the manuscript has gained in completeness, quality and clarity and we hope that this version is now acceptable for publication.

REVIEWER COMMENTS

Reviewer #1 (Remarks to the Author):

The manuscript "A paralog of Pcc1 is the fifth core subunit of KEOPS complex in Archaea" describes first in depth biochemical and structural characterization of an additional Pcc1-like subunit of essential in all three domains of life tRNA modification multisubunit enzyme called KEOPS. This is a very thorough, high-quality work. All key conclusions are strongly supported, and the text is clearly written. I also find convincing the presented evidence that the fifth eukaryotic subunit of KEOPS complex GON1 is a very diverged homolog of the Pcc1-like protein (or Pcc2 as it was named by authors), despite absence of easily detectable sequence and structural similarity. I have therefore only minimal criticism concerning some computational analyses and description of some computational methods.

Minor issues for correction/modification

1. Please provide details for the procedure of discrimination between Pcc1 and Pcc2 families and additional evidence supporting data shown in the Figure 2A. There are three options 1) show a phylogenetic tree with two respective branches indicated and protein ID and species name indicated for all leaves; 2) Alternatively, a complete multiple alignment with differentiating box2 shown can be provided; 3) For all proteins scores for two Pcc1 and Pcc2 HMM profiles and desirably a signature/sequence for box 2 can be provided in a Supplementary Table. For phylogenetic analysis it is OK to make non-redundant set using some objective criteria (eg. one representative for a cluster of 90% identical proteins) and remove fragments, but otherwise all proteins from the two families identified in all archaeal genomes should be included. In particular I am concerned by results for Crenarchaea where Pcc1 subunit is found in much smaller number of genomes than Pcc2. If this is indeed so, at least a note or, better, an explanation/discussion in the text should be included. Also please indicate an overall number of archaeal genomes and specify the database used to make the analysis shown in this figure. And provide HMM profiles or respective multiple alignments if they indeed discriminate these two families well.

We thank the reviewer for pointing this out. We have opted for the suggestion nr. 3 which we think allows to present in exhaustive and accessible manner the data that we used to prepare the part A of the figure 2. We added a supplementary material excel file which contains the raw data coming from HMM searches, the taxonomic distribution of hits, the E-values and accession number for identified Pcc1 and Pcc2 proteins. In addition, a manually curated summary of the results is given in a separate tab. The HMM profiles generated from this full set of sequences (see below) is very similar to that shown in the Figure 2B and Supplementary figure 3. We added a sentence in the legend of the supplementary figure 3 to indicate this. Finally, we added a methods section in the manuscript to explain how the sequence similarity searches were done.

Sequence logo generated from 1318 Pcc1 sequences:

Sequence logo generated from 1112 Pcc2 sequences:

Regarding the comment on Crenarchaeota : in addition to Crenarchaeota we identified less Pcc1 in five other lineages (as shown in Figure 2A). We don't have a conclusive answer as to why this is so especially in these lineages. We speculate that Pcc1 proteins may be particularly divergent in these organisms and therefore difficult to detect by sequence similarity searches or that the corresponding genes were not annotated as protein coding sequence by automatic annotation tools. Indeed, when searching for Pcc2 orthologs, in some instances we could identify hits only by performing manual tblastn searches with Pcc2 from the closest relative as a query sequence. So we think that the apparent absence of Pcc1 (and Pcc2) encoding genes in some genomes is probably the consequence of us being unable to detect them. We added this explanation in the results section.

2. Indicate how those 27 representatives were selected in either Methods section or all relevant Figure legends. Were they chosen manually?

The 27 sequences were chosen manually to cover the whole breadth of the archaeal phylogenetic tree. Specifically, sequences from all four superphyla were included proportionally to the number of phyla in each superphylum (see Supplementary Figure 3B for taxonomic assignment of sequences). When possible we chose species with complete genomes, otherwise MAGs were chosen for which the Pcc1 and Pcc2 encoding genes were found within the same MAG. We added a sentence in methods section to explain this.

3. The section on GON1 homologs in plants is beyond the scope of this work, I think. But if it is to stay, contamination of plant genomes by fungi contigs should be ruled out, because, unfortunately, this is the null hypothesis for this observation but not horizontal transfer from fungi to plants. To rule this out taxonomic affiliation of best hits for a dozen of neighboring proteins encoded up- and downstream from respective genes in two identified "plant" contigs should be shown in the

supplement. If results will be inconclusive, just include “or contamination” in the sentence in line 653.

Yes, we agree. We confirm that almost all neighbouring genes have closest hits with fungal sequences. We therefore added “or contamination” in line 653.

4. Resolution and/or font of several supplementary figures is very low (eg. SF2, SF3, SF9). Please fix this.

SF2: done

SF3: done

SF9: this is difficult, we don't see how we can increase the font without fragmenting too much the alignment (landscape format doesn't help). We could increase a little bit the font by drastically reducing the size of the descriptive labels next to each sequence (like we did in Figure 8B) but this would mean that the labels would not be self-explanatory anymore and we would need to add a long abbreviation list in the figure legend which we think does not make it easier to read the figure. It's not optimal, but on the computer screen the figure can be sufficiently magnified to be easy to read.

Minor and optional suggestions:

1. Pcc1 and Pcc2 are similar enough to be annotated in most databases as Pcc1. So, this is a trivial finding and does not require too much space to explain. It is just enough to say that most of archaeal genomes have two Pcc1-like proteins, but function is known only for one of them. Thus, consider streamlining this section of the text and removing Supplementary figure 1.

In our list of hits from the UniProt database (supplementary table file) only about 150 Pcc2 sequences were annotated as Pcc1 while the others (almost 1000) were classified as “uncharacterised proteins”. Even the majority of Pcc1 proteins (~900) in our hit list are referenced as “uncharacterised proteins”. So, it seems to us that detecting Pcc1 and Pcc2 orthologs is not as trivial as one would think. We were actually quite lucky to have noticed serendipitously the double Pcc1 annotation in the NCBI record of the *T. kodakarensis* genome. To illustrate this we performed search in the NCBI protein database with “Pcc1 Thermococcus kodakarensis” (see screenshot below) and this shows that while in *Thermococcus kodakarensis* the two proteins were annotated as Pcc1 in the strain KOD1 even the Pcc1 was left unannotated. We would therefore like to keep the Supplementary figure 1 and the text unchanged.

- KEOPS complex subunit Pcc1 [Thermococcus kodakarensis]
- 1. **81 aa protein**
 Accession: WP_173254056.1 GI: 1850591356
BioProject Nucleotide PubMed Taxonomy
GenPept Identical Proteins FASTA Graphics
- KEOPS complex subunit Pcc1 [Thermococcus kodakarensis]
- 2. **85 aa protein**
 Accession: WP_011250204.1 GI: 499569421
BioProject Nucleotide Taxonomy
GenPept Identical Proteins FASTA Graphics
- hypothetical protein, conserved [Thermococcus kodakarensis KOD1]
- 3. **85 aa protein**
 Accession: BAD85442.1 GI: 57159512
BioProject Nucleotide PubMed Taxonomy
GenPept Identical Proteins FASTA Graphics
- unnamed protein product [Thermococcus kodakarensis KOD1]
- 4. **85 aa protein**
 Accession: CAT69705.1 GI: 218092952
Nucleotide Taxonomy
GenPept Identical Proteins FASTA Graphics
- hypothetical protein, conserved [Thermococcus kodakarensis KOD1]
- 5. **82 aa protein**
 Accession: BAD84831.1 GI: 57158901
BioProject Nucleotide PubMed Taxonomy
GenPept Identical Proteins FASTA Graphics

2. Include a reference for the statement in line 100.

Done, we cited Mao et al., 2008 where the resemblance with KH domain was first detected.

3. Modify “small uncharacterized protein that showed significant sequence similarity” to “functionally uncharacterized homolog” (line 128), because technically most of them are annotated as Pcc1 in protein databases.

We would like to keep the phrasing as it is (see answer to point 1, minor comments).

4. Line 276. “was realized” correct to “was built”

done

5. Line 613. Please explain what “seemingly a different fold” means. A topology diagram would help here.

We have replaced this sentence by:

Consistent with this hypothesis, structural alignment of Pcc1/Pcc2 with human LAGE3/C14 complex shows a remarkable conservation of the interaction interface despite the fact that the b-sheets of C14/Gon7 and Pcc2 proteins have a different topology (b1b3b2 for Pcc2 and b1b2 for C14/Gon7) suggesting that this interface is a functionally important requirement that has been conserved from archaea to eukaryotes.

6. Line 673. Delete “,” after Tsab.

done

Reviewer #2 (Remarks to the Author):

A paralog of Pcc1 is the fifth core subunit of KEOPS complex in Archaea
Daugeron Marie-Claire^{1,§}, Missouri Sophia^{1,§,§}, Da Cunha Violette^{1,£}, Lazar Nouredine¹,
Collinet Bruno^{1,2}, van Tilbeurgh Herman^{1,#} & Tamara Basta¹

The authors describe the assignment of a 5th subunit in the archaeal form of the tRNA modifying enzyme complex KEOPS. They make a good case of the evolutionary relations between Archaea and Eukaryotes. The data they present is sound and convincing. They use the KEOPS genes from *P. abyssi* recombinantly expressed in *E. coli* as their model system. They clearly show that all 5 subunits form a functional complex and that the PCC1 and Pcc2 are responsible for the oligomeric states of the enzyme. The authors make a very interesting case that the 5 subunit KEOPS is an ancestral system, maybe part of the LUCA? I particularly like the implication this has for the ever expanding groups of Archaea.

Although I think the research is interesting and sound I have 10 major points to make. These need to be addressed in order to move forward in Nature Comm or other journal.

1-the title contains two abbreviations which I think is for a broad audience unacceptable. The tRNA modification function needs to be present in the title.

We think that writing out KEOPS (Kinase Endopeptidase and Other Proteins of Small size) would not be informative (rather the contrary, it would be confusing) for the broad audience and would make the title much longer. We agree however, that the tRNA modification function should appear in the title. The title was changed to:

“A paralog of Pcc1 is the fifth core subunit of the KEOPS tRNA modifying complex in Archaea”

2-the introduction does not give a quick overview of the bacterial system which does not utilize a 5 subunit complex. Please give a short description on the bacterial system to be used to discuss later in the evolutionary path of KEOPS.

Yes, we added a sentence in the introduction part to describe the bacterial system.

3-in the legend of figure 1 B it is not clear what is meant with the oligomeric state (also in the introduction). Where and when do you have physiologically active dimeric or monomeric complexes *in vivo* eg monomeric only active eukaryotes, Archaea both monomeric and dimeric, it gets confusing quickly?

We modified the legend of Figure 1B to clarify: “the oligomerization of the 4SU KEOPS complex into a dimer of two heterotetramers”

We also modified the Figure 1 and changed “oligomeric status” into “regulation of oligomerization” to be more explicit.

When and where physiologically active dimeric or monomeric complexes are present *in vivo* has not been specifically studied yet in eukaryotes and even less so in archaea. As we describe in the

introduction, so far, we know that KEOPS can be purified from yeast or human cells as a 5SU complex (heteropentamer). However, it is not known whether this complex co-exists with 4SU complex (which forms a dimer of two heterotetramers) in cells.

4-the authors use a short 80-90 residue protein for phylogenetic analysis, which I think is sound, but do not mention the difficulty which such short sequences brings in phylogenetic analysis.

Yes, agreed. The difficulty of working with such short sequences is that they contain only small amount of phylogenetically informative positions. In our case, there is fortunately enough signal to robustly segregate the two paralogs into two monophyletic groups (Figure 2C) but the signal is insufficient to resolve the topology of the tree within each clade. We now added a sentence in the legend of the figure 2 to explain this.

5-Figure2. Panel C is meaningless move to the supplement and please annotate a little bit. I like panel B but extend the structural top bit to the whole sequence and show the zoom in with the sequence logo. Move panel A to the supplement figure 2, I like supplement figure 2 much better but make it a little prettier, better to read and for god sake where are the Crenarchaeota?

We do not think that the panel C is meaningless and we would like to keep it. As stated above it shows that Pcc1 and Pcc2 sequences can be robustly segregated in two monophyletic clades, a topology that is expected for paralogous proteins. One could also imagine more complicated topologies (a part of Pcc2 sequences branching within Pcc1 clade for example) which would lead to very different evolutionary scenarios. It is therefore important to show, in our opinion, that such scenarios can be excluded.

Part B, we added the topology diagram for Pcc1 proteins on top of the two logos.

Part A, we agree that the supplementary figure 2 is easier to understand but it is also less informative. The names of different phyla are difficult to read even though we increased the font size as much as it was possible. Also, the reader does not have the information about the proportion of each paralog per phylum and could be misled to believe that Pcc1 and Pcc2 are systematically found in all representatives of a taxon. We therefore prefer to keep the table in the main figure and leave the tree, a more qualitative representation of the data, in the supplementary materials and methods.

Crenarchaeota order was split into several families (Sulfolobales, Thermoproteales, Desulfurococcales etc.) which is why it does not appear in the tree.

6-the authors really need to address the variation in the Archaea some have only Pcc1 some have 1 and 2 and some have neither, how does this work do they still have a KEOPS or do they have the bacterial system? Which becomes important in the evolutionary path in the discussion

As said in the answer to the reviewer 1 above (Point 1 major comments), the variation is likely due to our incapacity to detect all the orthologs by sequence similarity searches, some genes were missed by annotation tools and, finally, most of the genomes in the databases are incomplete.

7-Figure 3 they authors show a chromatogram trace of the gelfiltration column, almost like copied

and past out of the software that runs the equipment. That is just lazy and unacceptable, make a nice figure and put in the appropriate sizes obtained with the gelfiltration analysis.

We think that calling fellow colleagues “lazy” is unacceptable. In addition, we do not understand what the reviewer means by “nice figure”? The figure as it is, highlights the most informative part of the chromatogram. If the reader is curious, the complete chromatograms and the corresponding gels are available in the supplementary figure 4. To our opinion, it makes good sense to exploit the modern software that pilots the FPLC machine to prepare figures.

8-I got very confused by the SAXS data, how is correlated to the gelfiltration data, is it the same or different what is obtained from the SAXS data and gelfiltration, any problems with not homogenous samples for SAXS? A table with the different oligomeric states in the different complexes will be helpful.

The gel filtration experiments shown in Figure 3 were performed at high salt (500 mM NaCl, see material and methods) conditions to stabilise the complexes. However, under these conditions, it was not possible to accurately determine the molecular mass of each complex, we only could compare the elution profiles relative to each other. In the SEC-SAXS experiments, the complexes were first separated by gel filtration under low salt conditions (150 mM) and we did observe a partial disassembly of the KBCP1 and KBCP1P2 complexes in addition to a major peak corresponding to the whole complex. The major peaks were used to determine the molecular mass of each complex (from the elution volume) and to collect the SAXS data. The stoichiometry of each complex is indicated in Figure 6 and the complete summary of the collected SAXS data can be found in the Supplementary table 2.

9-Note added in proof? What proof? this is the first version of the manuscript. I you point my attention to a manuscript in a preprint server you will have to discuss it. You cannot just mention it as a quick note that is unacceptable (I do not like preprint servers either but you have no choice in that). Please address this preprint manuscript properly as it significantly overlaps with this work and might even strengthen it.

Yes, we agree. We deleted “note in proof” and addressed properly the preprint article from Wu and colleagues in the discussion section.

10-The authors have not convinced me of the necessity of dimerization prevention in Archaea (Eukaryotes are clearer in that) as described in the conclusion. This is especially confusing as not all Archaea seem to have Pcc2 please enlighten the reader on that.

Yes we agree, with the current state of the knowledge (see our answer to the reviewer’s comment 3) it is inappropriate to use the term “necessity” we therefore changed to “possibility”.

Reviewer #3 (Remarks to the Author):

The study by Daugeron and coworkers describes the identification and biochemical and structural characterization of a novel archaeal auxiliary protein factor (Pcc2) that is part of the KEOPS complex.

The KEOPS complex catalyzes the second step of the t6A modification at position 37 of most tRNAs with UNN anticodons, an essential and universal post-transcriptional modification. The archaeal KEOPS complex was thought to consist of 4 distinct protein components (Kae1, Bud32, Pcc1, and Cgi121), contrasting with the eukaryotic complex that contains 5. In this work, the authors used co-purification and biophysical (X-ray crystallography and anisotropy) approaches to show that Pcc2, a paralog of Pcc1, may bind to the KEOPS complex by directly interacting with either the Pcc1 or the Kae1 subunit (the catalytic site). The crystal structures of Pcc2 and the Pcc2-Pcc1 complex were solved, providing detailed insights into the interactions of these factors. SAXS-based structural models of the

KEOPS complex in the presence and absence of Pcc2 or Pcc1 suggest that Pcc2 prevents the dimerization of the KEOPS complex and illustrate how it may function. While the presence of Pcc2 decreases the tRNA binding affinity, it does not significantly affect the activity of the KEOPS complex. Notably, the absence of Pcc2 does not impact the KEOPS complex's activity.

Overall, this work could represent an exciting discovery of interest for the field that may be of interest to a broad readership. The manuscript is well-written, and the data generally support the conclusions. However, there are a few significant gaps in the study regarding the biological aspects of the work, which detracts from its significance. A major gap is that Pcc2 was shown to bind to the KEOPS complex when over-expressed heterologously in *E. coli*; it is unclear whether this is a bona fide interaction in the host. This is important because Pcc2 may not be an exclusive component of the KEOPS complex as it may have alternative functions and interacting partners. This could reconcile the fact that Pcc2 is not necessary for the t6A modification *in vitro* but is essential for cell viability, as discussed by the authors. Thus, this interaction should be validated in the host species. Another critical gap is that there needs to be a robust discussion about the biological function of Pcc2. The authors pose that Pcc2 evolved to prevent dimerization of the KEOPS complex, but no explanation of the biological relevance is provided. While the biochemical and structural data presented are interesting and provide mechanistic insights into how the complex may operate, information on the biological importance of Pcc2 is needed, which would substantially strengthen the significance of this work. Additional recommendations to improve this study are outlined below.

- Important controls are missing in the co-purification experiments to prove that the binding of Pcc2 is specific. For example, can Pcc2 co-purify with a complex missing Kae1?

If we understand correctly, the referee wants to know if Pcc2 can bind to Bud32 or Cgi121 in the absence of Kae1. We could in principle test this but we are not sure how relevant this is given that such situation does not occur in the cell. Kae1 protein is essential and therefore has to be always present in some minimal quantity in the cell. If the question is whether Pcc2 is a "sticky" protein, the answer is no: during our KEOPS purification procedure we reconstitute the whole complex from subcomplexes and in some cases Pcc2 is in large excess over the other subunits. Despite this, we never observed peaks corresponding to Bud32-Pcc2 or Cgi-Pcc2 complexes or Pcc2-*E. coli* proteins in the gel filtration profiles.

- Based on their phylogenetic analyses, the authors propose that Pcc2 and Gon7 are functionally related homologs. This hypothesis could be tested using their established co-purification binding assays. Can Gon7/C14 replace Pcc2?

Yes, we would love to know if Pcc2 and Gon7 are functional homologs. However, testing whether the two proteins interact *in vitro* would not truly answer this question. At best, we could show that the

two proteins co-purify and that Gon7 engages archaeal Pcc1 via the dimerization interface (for that we would need the crystal structure of the binary Pcc1-Gon7 complex). Given the strong structural resemblance between archaeal and yeast Pcc1 this would be expected. Actually, we could model such a complex by using the AlphaFold2 and the predicted interface (composed of 2 alpha helices plus 5 stranded antiparallel beta sheet) is identical to that of the yeast Pcc1-Gon7 complex:

Perhaps a more meaningful way to test functional homology would be to establish an *in vivo* complementation test whereby we would replace Pcc2 by Gon7 or C14. We can not do this in *Thermococcus* (there are no thermostable Gon7/C14 proteins) but we could, in principle, test it in yeast by replacing Gon7 by an archaeal Pcc2. Since Gon7 knock out mutants are slow growing we could test if Pcc2 could rescue this phenotype. Unfortunately, our previous experience shows that the chances are high that such an experiment would be inconclusive. Indeed, we were unable to complement Gon7 deletion mutant with the human ortholog C14 although there is no doubt that the two are functional homologs (Braun et al., 2017, Nat. Gen. 49: 1529-1538, <https://doi.org/10.1038/ng.3933>).

We think that the way to go to answer the question of functional homology would be to characterise the *in vivo* function of Pcc2 and Gon7/C14 proteins but this will require a study of its own.

•Can the authors comment on the potential biological role of Pcc2? Is it a regulatory factor? Why is it important for the Pcc2 to prevent KEOPS superdimers? Is it co-expressed with the other KEOPS factors in the host organism? Could it play a role in the first step of the modification reaction?

These are all good questions which we tried to address in the discussion part of the manuscript (Pcc2 is essential in archaea). With the lack of more *in vivo* data we can only speculate on the basis of our *in vitro* data and the comparison with what is known about Gon7/C14 proteins. Since monomeric and dimeric KEOPS complexes have comparable tRNA modification activity *in vitro* (both in our hands and in human system), the role of this dimerization remains unclear. In human cells, C14 enhances or stabilises the expression levels of KEOPS since in the absence of C14 the global levels of KEOPS subunits (and t⁶A levels) is significantly lower perhaps explaining why Gon7/C14 deletion has deleterious effect on yeast and human cells (Arrondel et al. <https://doi.org/10.1038/s41467-019-11951-x>). We see a reminiscent behaviour *in vitro* whereby the 4SU KEOPS complex has a tendency to form aggregates suggesting that Pcc2 could play a role in stabilising the complex *in vivo*.

Regarding the co-expression question: we did not perform an exhaustive bioinformatics analyses but, so far, we could not detect operons containing pcc2 and other KEOPS subunits. As a rule, we don't

find KEOPS subunits as part of operons with exception of *bud32* and *kae1* genes that form operons in some archaeal species.

Regarding the first step of the reaction question: we think it is highly unlikely that Pcc2 could play a direct role in the first part of the reaction. This reaction is catalysed by a stand-alone enzyme Sua5/TsaC and it was shown (in vitro though) that a direct contact between Sua5/TsaC and KEOPS complex is not necessary for the t⁶A synthesis. However, it was postulated on the basis of structural data and enzyme kinetics data (Lauhon et al., DOI: 10.1021/bi301233d) that the reaction intermediate TC-AMP is channelled from the active site of Sua5/TsaC to the active site of Kae1. If so, a direct interaction between Sua5 and KEOPS complex needs to be established at least transiently. Perhaps such interaction is easier to establish with 50S KEOPS complex, however, at least in vitro, this seems not to have a significant effect on the rate of t⁶A synthesis. Interestingly, in their preprint article (<http://biorxiv.org/lookup/doi/10.1101/2022.08.26.505501>) Wu and colleagues report the detection of Sua5 when affinity purifying KEOPS proteins from *Saccharolobus islandicus* cell extracts. It may therefore be possible to test in the future the effect of Pcc2 on the interaction between Sua5 and KEOPS in this organism.

- Overall, the methodology section provides enough details to reproduce the experiments. However, the sequence of the DNA template (including the ribozyme) for the in vitro transcription of the tRNA substrate should be included in the manuscript.

We added the complete sequence of the DNA template in the materials and methods section.

- Can the authors comment on the evolutionary implications of the absence of Pcc2 in some species?

Yes, as explained above (see comment 1 reviewer 1) it is likely that the apparent absence of Pcc2 in some species is due to our incapacity to detect them by sequence similarity searches and the fact that many genomes in the databases are incomplete. This being said, we noticed that in the DPANN superphylum we could identify Pcc2 in only one phylum out of nine. DPANN archaea were missed in the initial metagenomics studies because they have very divergent 16S rDNA sequences. It turns out that many of these organisms are nano-sized and usually harbour small genomes suggesting that they could be parasites or symbionts. It is likely that these organisms are undergoing reductive genome evolution and are losing many genes. It makes sense to think that, as a part of that evolutionary process, Pcc2 could be easier to lose than, say, the catalytic subunit Kae1, since Pcc2 seems not to impact directly the t⁶A synthesis.

Reviewer #1 (Remarks to the Author):

Although I find response to my concerns satisfactory, in general, I still believe that more clarity is required with respect of assignments of proteins to Pcc1 or Pcc2 groups.

>After two iterations, we recovered 1318 Pcc1 sequences 617 and 1112 Pcc2 sequences that were all from archaea

1. Could you please provide more details here, because, I assume, two iterations of the Jackhammer identify both Pcc1 and Pcc2. In other words, if you start with Pcc1 alignment you will find Pcc2 proteins and vice versa. So I would like to see results (E-values) of these searches with two profiles included in the "all hits" table for each accession. Now it is just one E-value provided (column E) and it is unclear which profile gives the hit with respective E-value. Moreover, as far as I can tell at least some of those are from the first, but not the second iteration. Please check.

2. Furthermore, I found several duplicates (two lines for the same accession) with different assignments:

A0A101DKX8_9EURY Thermococcales archaeon 44_46 Pcc1
A0A101DKX8_9EURY Thermococcales archaeon 44_46 Pcc2
A0A151BBS7_9ARCH Candidatus Bathyarchaeota archaeon B24 Pcc1
A0A151BBS7_9ARCH Candidatus Bathyarchaeota archaeon B24 Pcc2
A0A151BHF6_9ARCH Candidatus Bathyarchaeota archaeon B23 Pcc1
A0A151BHF6_9ARCH Candidatus Bathyarchaeota archaeon B23 Pcc2
A0A151E681_9EURY Thermoplasmatales archaeon SG8-52-2 Pcc1
A0A151E681_9EURY Thermoplasmatales archaeon SG8-52-2 Pcc2
A0A151EQF6_9EURY Thermoplasmatales archaeon SG8-52-3 Pcc1
A0A151EQF6_9EURY Thermoplasmatales archaeon SG8-52-3 Pcc2
A0A1D2R4Q9_9EURY Candidatus Altiarchaeales archaeon WOR_SM1_79 Pcc1a
A0A1D2R4Q9_9EURY Candidatus Altiarchaeales archaeon WOR_SM1_79 Pcc2
M7TAP7_9EURY Thermoplasmatales archaeon SCGC AB-539-N05 Pcc1
M7TAP7_9EURY Thermoplasmatales archaeon SCGC AB-539-N05 Pcc2

So please double check and/or explain how this is possible.

3. Also, I cannot find any data from Jackhammer searches for, eg. A0A2J6N8P2_9ARCH proteins, which is assigned to Pcc2 group. How was it done then? Please check other such cases.

4. Update methods accordingly and correct Figure 2A if needed.

Minor:

>Line 194: Tree scale gives the correspondence between the branch length and sequence evolution.

In which units? In substitution per site?

Reviewer #2 (Remarks to the Author):

I think the authors did a good job in adjusting the manuscript in response to the reviewer comments. The small change to the title is perfect. In my view it is ready for the next step of the publication procedure.

Reviewer #3 (Remarks to the Author):

The authors have addressed most of my comments satisfactorily.

Reviewer #1 (Remarks to the Author):

Although I find response to my concerns satisfactory, in general, I still believe that more clarity is required with respect of assignments of proteins to Pcc1 or Pcc2 groups.

>After two iterations, we recovered 1318 Pcc1 sequences 617 and 1112 Pcc2 sequences that were all from archaea

1. Could you please provide more details here, because, I assume, two iterations of the Jackhmmmer identify both Pcc1 and Pcc2. In other words, if you start with Pcc1 alignment you will find Pcc2 proteins and vice versa. So I would like to see results (E-values) of these searches with two profiles included in the “all hits” table for each accession. Now it is just one E-value provided (column E) and it is unclear which profile gives the hit with respective E-value. Moreover, as far as I can tell at least some of those are from the first, but not the second iteration. Please check.

Yes, thank you, the sentence is actually wrong. We corrected it and additionally provided more detailed account about the results of the iterative Jackhmmmer search (see point 4 below). As requested by the reviewer, we included the e-values in the “all hits” table for searches with two profiles. This shows that the two paralogs can be stringently distinguished from each other based on the search we performed. In the majority of cases, the search is exclusive. In some cases, we do detect Pcc2 orthologs when searching with Pcc1 sequences and vice versa but we can easily identify the genuine hit because it has orders of magnitude lower e-value as compared to the contaminating sequence.

2. Furthermore, I found several duplicates (two lines for the same accession) with different assignments:

A0A101DKX8_9EURY Thermococcales archaeon 44_46 Pcc1
A0A101DKX8_9EURY Thermococcales archaeon 44_46 Pcc2
A0A151BBS7_9ARCH Candidatus Bathyarchaeota archaeon B24 Pcc1
A0A151BBS7_9ARCH Candidatus Bathyarchaeota archaeon B24 Pcc2
A0A151BHF6_9ARCH Candidatus Bathyarchaeota archaeon B23 Pcc1
A0A151BHF6_9ARCH Candidatus Bathyarchaeota archaeon B23 Pcc2
A0A151E681_9EURY Thermoplasmatales archaeon SG8-52-2 Pcc1
A0A151E681_9EURY Thermoplasmatales archaeon SG8-52-2 Pcc2
A0A151EQF6_9EURY Thermoplasmatales archaeon SG8-52-3 Pcc1
A0A151EQF6_9EURY Thermoplasmatales archaeon SG8-52-3 Pcc2
A0A1D2R4Q9_9EURY Candidatus Altiarchaeales archaeon WOR_SM1_79 Pcc1a
A0A1D2R4Q9_9EURY Candidatus Altiarchaeales archaeon WOR_SM1_79 Pcc2
M7TAP7_9EURY Thermoplasmatales archaeon SCGC AB-539-N05 Pcc1
M7TAP7_9EURY Thermoplasmatales archaeon SCGC AB-539-N05 Pcc2

So please double check and/or explain how this is possible.

Yes, this is a mistake. These are either Pcc1 or Pcc2 but not both. We corrected this and checked for additional duplicates but found none.

3. Also, I cannot find any data from Jackhmmmer searches for, eg. A0A2J6N8P2_9ARCH proteins, which is assigned to Pcc2 group. How was it done then? Please check other such cases.

We checked, there is 7 such cases. We can't explain why there is no data from Jackhmmmer search for these proteins. We checked them individually (by doing global pairwise alignment against Pcc1 or Pcc2 of *T. kodakarensis*) and could assign them as Pcc1 or Pcc2 based on the quality of the alignment and the presence of SN or DD motif. We indicated this in the "all hits" table.

4. Update methods accordingly and correct Figure 2A if needed.

We corrected Figure 2A and we updated methods as following:

To search for Pcc1 and Pcc2 orthologs, we performed iterative searches against UniProtKB database using the Jackhmmr program <https://www.ebi.ac.uk/Tools/hmmer/search/jackhmmmer> with default settings. We manually selected 21 representative archaeal species each encoding both Pcc1 and Pcc2. We then generated separate alignments for Pcc1 or Pcc2 sequences using MAFFT (43) and used those as input. After two iterations, we retrieved 1264 significant hits that could be unambiguously assigned as archaeal Pcc1 (646 proteins) or Pcc2 (618 proteins) based on e-values. The table containing the raw data and the summary of the search results is available as supplementary file.

Minor:

>Line 194: Tree scale gives the correspondence between the branch length and sequence evolution.

In which units? In substitution per site?

Yes, we corrected the legend of the Figure 2 correspondingly.

Reviewer #2 (Remarks to the Author):

I think the authors did a good job in adjusting the manuscript in response to the reviewer comments. The small change to the title is perfect. In my view it is ready for the next step of the publication procedure.

Reviewer #3 (Remarks to the Author):

The authors have addressed most of my comments satisfactorily.